# Zipper: Addressing Degeneracy in Algorithm-Agnostic Inference

**Geng Chen**[*] **Yinxu Jia**[*] **Guanghui Wang**[*†] **Changliang Zou**[*†]

NITFID, School of Statistics and Data Science, LPMC, KLMDASR, and LEBPS, Nankai University

gengchen.stat@gmail.com, yxjia@mail.nankai.edu.cn,
ghwang.nk@gmail.com, zoucl@nankai.edu.cn

## Abstract

The widespread use of black box prediction methods has sparked an increasing interest in algorithm/model-agnostic approaches for quantifying goodness-of-fit, with direct ties to specification testing, model selection and variable importance assessment. A commonly used framework involves defining a predictiveness criterion, applying a cross-fitting procedure to estimate the predictiveness, and utilizing the difference in estimated predictiveness between two models as the test statistic. However, even after standardization, the test statistic typically fails to converge to a non-degenerate distribution under the null hypothesis of equal goodness, leading to what is known as the degeneracy issue. To addresses this degeneracy issue, we present a simple yet effective device, Zipper. It draws inspiration from the strategy of additional splitting of testing data, but encourages an overlap between two testing data splits in predictiveness evaluation. Zipper binds together the two overlapping splits using a slider parameter that controls the proportion of overlap. Our proposed test statistic follows an asymptotically normal distribution under the null hypothesis for any fixed slider value, guaranteeing valid size control while enhancing power by effective data reuse. Finite-sample experiments demonstrate that our procedure, with a simple choice of the slider, works well across a wide range of settings.

## 1 Introduction

Consider predicting response $Y \in \mathbb{R}$ from covariates $X \in \mathbb{R}^p$. Due to the popularity of black box prediction methods like random forests and deep neural networks, there has been a growing interest in the so-called "*algorithm (or model)-agnostic*" inference on the *goodness-of-fit* (GoF) in regression [1, 2, 3, 4, 5]. This framework aims to assess the appropriateness of a given model for prediction compared to a potentially more complex (often higher-dimensional) model. A common approach involves defining a predictiveness criterion, employing either the *sample-splitting* or *cross-fitting* strategy to estimate the predictiveness of the two models, and examining the difference in predictiveness. The main focus of this work is to address the issue of degeneracy encountered in predictiveness-comparison-based test statistics.

### 1.1 Goodness-of-Fit Testing via Predictiveness Comparison

Let $P$ represent the joint distribution of $(Y, X)$ and consider a class $\mathcal{F}$ of prediction functions that effectively map the covariates to the response. Define a criterion $\mathbb{C}(\tilde{f}, P)$, which quantifies the predictive capability of a prediction function $\tilde{f} \in \mathcal{F}$. Larger values of $\mathbb{C}(\tilde{f}, P)$ indicate stronger

---

[*]All authors contributed equally to this work and are listed in alphabetical order.

[†]Guanghui Wang and Changliang Zou are corresponding authors.

38th Conference on Neural Information Processing Systems (NeurIPS 2024).

predictive capability. The optimal prediction function within the class $\mathcal{F}$ is determined as $f \in \arg\max_{\tilde{f} \in \mathcal{F}} \mathbb{C}(\tilde{f}, P)$. Therefore, $\mathbb{C}(f, P)$ represents the highest achievable level of predictiveness within the class $\mathcal{F}$. Examples of $\mathbb{C}$ include the (negative) squared loss $\mathbb{C}(\tilde{f}, P) = -E[\{Y - \tilde{f}(X)\}^2]$ for continuous responses and the (negative) cross-entropy loss $\mathbb{C}(\tilde{f}, P) = E[Y \log \tilde{f}(X) + (1 - Y) \log\{1 - \tilde{f}(X)\}]$ for binary responses, where $E$ denotes the expectation under $P$.

In GoF testing problems, there are typically two classes of prediction functions $\mathcal{F}$ and $\mathcal{F}_{\mathcal{S}}$, where $\mathcal{F}_{\mathcal{S}}$ is a subset of $\mathcal{F}$. Define a dissimilarity measure $\psi_{\mathcal{S}} = \mathbb{C}(f, P) - \mathbb{C}(f_{\mathcal{S}}, P)$, which quantifies the deterioration in predictive capability resulting from constraining the model class to $\mathcal{F}_{\mathcal{S}}$. Here, $f_{\mathcal{S}} \in \arg\max_{\tilde{f} \in \mathcal{F}_{\mathcal{S}}} \mathbb{C}(\tilde{f}, P)$ denotes the optimal prediction function within the restricted class. Since $\mathcal{F}_{\mathcal{S}} \subseteq \mathcal{F}$, it follows that $\psi_{\mathcal{S}} \geq 0$. The GoF testing is then formulated as

$$H_0 : \psi_{\mathcal{S}} = 0 \text{ versus } H_1 : \psi_{\mathcal{S}} > 0. \tag{1}$$

The formulation of assessing a scalar predictiveness quantity is inspired by the work of Williamson et al. [5], which focuses on evaluating variable importance. Other predictiveness or risk quantities have also been explored by [6, 7, 8, 9]. Here, we highlight a few examples where the aforementioned framework can be directly applied.

- (Specification testing) Model specification testing aims to evaluate the adequacy of a class of postulated models, such as parametric models, say, examining whether the conditional expectation $E(Y \mid X) = g_\theta(X)$ holds almost surely [10], where $g_\theta$ is a known function up to an unknown parameter $\theta$. Under the framework (1), we can consider $\mathcal{F}$ as a generally unrestricted class, while $\mathcal{F}_{\mathcal{S}}$ represents a class of parametric models.

- (Model selection) GoF testing can also be employed to identify the superior predictive model from a set of candidates [11, 12]. This situation often arises when choosing between two prediction strategies, such as an unregularized model and a regularized one. Testing $H_0$ is to assess whether the inclusion of a regularizer provides benefits for predictions.

- (Variable importance measure) A recently popular aspect of GoF testing involves evaluating the significance of a specific group of covariates $U$ in predicting the response, where $X = (U^\top, V^\top)^\top$. This assessment can be incorporated into (1) by defining a subset of prediction functions $\mathcal{F}_{\mathcal{S}} \subseteq \mathcal{F}$ that disregard the covariate group $U$ when making predictions. When utilizing the squared loss, $\psi_{\mathcal{S}}$ simplifies to the LOCO (Leave Out COvariates) variable importance measure [1, 13, 14, 5, 15, 16], which quantifies the increase in prediction error resulting from the removal of $U$. Specifically, taking $\psi_{\mathcal{S}} = E[\{Y - E(Y \mid V)\}^2] - E[\{Y - E(Y \mid X)\}^2]$ corresponds to testing for *conditional mean independence* [17, 18].

The measure $\psi_{\mathcal{S}}$ possesses the advantages of being both model-free and algorithm-agnostic. It is not tied to a specific model and remains independent of any particular model fitting algorithm. This flexibility makes $\psi_{\mathcal{S}}$ a versatile quantity for evaluating goodness-of-fit, enabling us to leverage diverse machine learning prediction algorithms in its estimation.

## 1.2 The Degeneracy Issue

Let $Z = (Y, X) \sim P$, and suppose we have collected a set of independent realizations of $Z$ as $Z_i = (Y_i, X_i)$ for $i = 1, \ldots, n$. To effectively estimate $\psi_{\mathcal{S}}$ while ensuring algorithm-agnostic inference, the sample-splitting or cross-fitting has recently gained significant popularity. This approach relaxes the requirements imposed on estimation algorithms, allowing for the utilization of flexible machine learning techniques [13, 19, 2, 20, 21]. Taking sample-splitting as an example, the data is divided into a training set and a testing set. Based on the training data, estimators $f_n^{\text{tr}}$ and $f_{n,\mathcal{S}}^{\text{tr}}$ are obtained for the optimal prediction functions $f$ and $f_{\mathcal{S}}$, respectively. The dissimilarity measure $\psi_{\mathcal{S}}$ can then be evaluated using the testing data, i.e., $\psi_{n,\mathcal{S}} = \mathbb{C}(f_n^{\text{tr}}, P_n^{\text{te}}) - \mathbb{C}(f_{n,\mathcal{S}}^{\text{tr}}, P_n^{\text{te}})$, where $P_n^{\text{te}}$ represents the empirical distribution function of the testing data. Notably, in the context of measuring variable importance, it has been established that when $\psi_{\mathcal{S}} > 0$, the estimator $\psi_{n,\mathcal{S}}$ exhibits *asymptotic linearity* under certain assumptions [5]. Similar results are also applicable to the problem of comparing multiple algorithms or models [11, 12].

However, the situation becomes distinct when considering the null hypothesis $H_0 : \psi_{\mathcal{S}} = 0$. Recent studies have drawn significant attention to a phenomenon known as *degeneracy* [22, 23, 15, 18, 5].

Consider the simplest scenario where there are no covariates and the objective is to test whether $\mu := E(Y) = 0$; see also Example 1 in Lei [11]. In this case, we set $\mathcal{F} = \mathbb{R}$ and $\mathcal{F}_{\mathcal{S}} = \{0\}$. Using the squared loss, we have $\psi_{\mathcal{S}} = E(Y^2) - E\{(Y - \mu)^2\} = \mu^2$. The estimator for $\psi_{\mathcal{S}}$ based on sample-splitting is $\psi_{n,\mathcal{S}} = 2\bar{Y}_n^{\text{te}}\bar{Y}_n^{\text{tr}} - (\bar{Y}_n^{\text{tr}})^2$, where $\bar{Y}_n^{\text{tr}}$ and $\bar{Y}_n^{\text{te}}$ denote the sample means of the training and testing data, respectively. When $\mu \neq 0$, $\sqrt{n}(\psi_{n,\mathcal{S}} - \mu)$ follows an asymptotic normal distribution. However, when $\mu = 0$, $\sqrt{n}\psi_{n,\mathcal{S}} = O_P(n^{-1/2})$, indicating the presence of the degeneracy phenomenon. While inference at a $n$-rate remains feasible under degeneracy in this specific example, it is crucial to recognize that this would not hold true for intricate models and black box fitting algorithms. Williamson et al. [5] provide evidence for the occurrence of degeneracy in a general variable importance measure $\psi_{\mathcal{S}}$, where the influence function becomes exactly zero under the null hypothesis. It is therefore required to address this degeneracy issue in a generic manner to perform algorithm-agnostic statistical inference.

## 1.3  Existing Solutions

By using the fact that the influence functions of the individual components $\mathbb{C}(f, P)$ and $\mathbb{C}(f_{\mathcal{S}}, P)$ in $\psi_{\mathcal{S}}$ remain nondegenerate even under $H_0$, Williamson et al. [5] proposed an additional data split of the testing data, where the two predictiveness functions are evaluated on two separate testing data splits. This approach ensures a nondegenerate influence function under $H_0$, therefore restoring asymptotic normality. A similar approach has been independently explored by Dai et al. [23]. However, performing additional data splits may reduce the actual sample size used in the testing, leading to a substantial loss of power. Alternatively, Rinaldo et al. [13] and Dai et al. [23] introduced data perturbation methods, where independent zero-mean noises are added to the empirical influence functions. These methods also restore asymptotic normality. However, determining the appropriate amount of perturbation to achieve desirable Type I error control remains a heuristic process. Furthermore, Verdinelli and Wasserman [22] proposed expanding the standard error of the estimator to mitigate the impact of degeneracy.

## 1.4  Our Contributions

In this paper, we introduce the Zipper device for algorithm-agnostic inference under the null hypothesis $H_0$ of equal goodness. Our approach is inspired by the method of additional splitting of testing data, as demonstrated in the works of Williamson et al. [5] and Dai et al. [23] for assessing variables with zero-importance. Instead of creating two distinct testing data splits to evaluate the discrepancy of predictiveness criteria between the expansive and restricted models, we encourage an overlap between them. The Zipper device serves to bind together the two overlapping splits, with a *slider* parameter $\tau \in [0, 1)$ controlling the proportion of overlap. To ensure stable inference and accommodate versatile machine learning prediction algorithms, we incorporate a $K$-fold cross-fitting scheme. We will demonstrate that the proposed test statistic follows an asymptotically normal distribution $H_0$ for any fixed value of $\tau$, ensuring valid size control while providing satisfactory power enhancement.

## 1.5  Related Works

For variable importance assessments, in addition to LOCO methods within our framework, Shapley value-based measures are commonly used [24, 25, 26, 27]. These measures, which estimate the incremental predictive accuracy contributed by a specific variable across all possible covariate subsets, reveal complex inter-variable relationships but at a considerable computational expense. Furthermore, conditional randomization tests [28, 29] offer a robust alternative when covariate distributions are known or can be accurately estimated. These methods are especially beneficial in semi-supervised settings with extensive unlabeled data. Additionally, LIME [30] focuses on estimating variable importance locally for a specific instance, while LOCO methods are designed to assess global variable importance.

## 1.6  Organization

The remainder of our paper is structured as follows. In Section 2, we introduce the Zipper device for addressing the degenerate issue, and present the asymptotic behaviors of the method. Finite-sample experiments are presented in Section 3. Section 4 concludes the paper. Proofs of theorems and additional numerical results are provided in Appendix.

## 2 Our Remedy

### 2.1 The Zipper Device

To initiate the process, we randomly partition the data into $K$ folds, denoted as $\mathcal{D}_1, \ldots, \mathcal{D}_K$, ensuring that each fold is approximately of equal size. For a given fold index $k \in \{1, \ldots, K\}$, we construct estimators $f_{k,n}$ and $f_{k,n,\mathcal{S}}$ for the oracle prediction functions $f$ and $f_{\mathcal{S}}$ correspondingly, using the data excluding the fold $\mathcal{D}_k$. To activate the Zipper device, we further randomly divide $\mathcal{D}_k$ into two splits, labeled as $\mathcal{D}_{k,A}$ and $\mathcal{D}_{k,B}$, with approximately equal sizes, allowing for an overlap denoted as $\mathcal{D}_{k,o}$. Specifically, $\mathcal{D}_{k,A} \cup \mathcal{D}_{k,B} = \mathcal{D}_k$ and $\mathcal{D}_{k,A} \cap \mathcal{D}_{k,B} = \mathcal{D}_{k,o}$. Let $\mathcal{D}_{k,a} = \mathcal{D}_{k,A} \backslash \mathcal{D}_{k,o}$ and $\mathcal{D}_{k,b} = \mathcal{D}_{k,B} \backslash \mathcal{D}_{k,o}$ represent the non-overlapping parts. For simplicity, we assume that $|\mathcal{D}_k| = n/K$ and $|\mathcal{D}_{k,A}| = |\mathcal{D}_{k,B}|$, where $|\mathcal{A}|$ represents the cardinality of set $\mathcal{A}$. Let $\tau = |\mathcal{D}_{k,o}|/|\mathcal{D}_{k,A}|$ represent the proportion of the overlap. Visualize the two splits $\mathcal{D}_{k,A}$ and $\mathcal{D}_{k,B}$ as two pieces of fabric or other materials. The term "Zipper" is derived from the analogy of using a zipper mechanism to either separate or join them by moving the slider $\tau$; see Figure 1. To evaluate the discrepancy measure $\psi_{\mathcal{S}} = \mathbb{C}(f, P) - \mathbb{C}(f_{\mathcal{S}}, P)$ based on the $k$th fold of testing data $\mathcal{D}_k$, we denote $P_{k,n,I}$ as the empirical distribution of the data split $\mathcal{D}_{k,I}$, where $I \in \{o, a, b\}$. We can estimate $\psi_{\mathcal{S}}$ by $\mathbb{C}_{k,n} - \mathbb{C}_{k,n,\mathcal{S}}$, where

$$\mathbb{C}_{k,n} = \tau \mathbb{C}(f_{k,n}, P_{k,n,o}) + (1-\tau)\mathbb{C}(f_{k,n}, P_{k,n,a}) \text{ and}$$
$$\mathbb{C}_{k,n,\mathcal{S}} = \tau \mathbb{C}(f_{k,n,\mathcal{S}}, P_{k,n,o}) + (1-\tau)\mathbb{C}(f_{k,n,\mathcal{S}}, P_{k,n,b})$$

represent weighted aggregations of empirical predictiveness criteria corresponding to overlapping and non-overlapping parts, used for estimating $\mathbb{C}(f, P)$ and $\mathbb{C}(f_{\mathcal{S}}, P)$, respectively. By employing the cross-fitting process, we obtain the final estimator of $\psi_{\mathcal{S}}$, denoted as

$$\psi_{n,\mathcal{S}} = K^{-1} \sum_{k=1}^{K} (\mathbb{C}_{k,n} - \mathbb{C}_{k,n,\mathcal{S}}), \tag{2}$$

which is the average over all testing folds.

Notably, if we fully open the Zipper, setting $\tau = 0$ and leaving $\mathcal{D}_{k,o}$ empty, our approach aligns with the vanilla data splitting strategy utilized in Williamson et al. [5] and Dai et al. [23] for assessing variables with zero-importance. Conversely, when we completely close Zipper with $\tau = 1$, our method essentially involves evaluating the predictiveness discrepancy using the identical data split $\mathcal{D}_{k,o} = \mathcal{D}_{k,A} = \mathcal{D}_{k,B} = \mathcal{D}_k$, which is known to result in the phenomenon of degeneracy [5]. Therefore, to avoid such degeneracy, we restrict the slider parameter $\tau \in [0, 1)$; see also Remark 2.5 below.

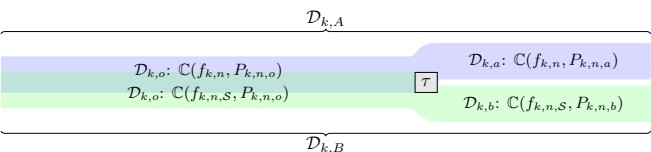

Figure 1: Illustration of the mechanism of the Zipper device based on the $k$th fold of testing data.

### 2.2 Asymptotic Linearity

We start by investigating the asymptotic linearity of the proposed test statistic $\psi_{n,\mathcal{S}}$ in (2), which serves as a foundation for establishing the asymptotic distribution under the null hypothesis, as well as for analyzing the test's power.

**Theorem 2.1** (Asymptotic linearity). *If Conditions (C1)–(C4) in Section A hold for both tuples* $(P, \mathcal{F}, f, f_{k,n})$ *and* $(P, \mathcal{F}_{\mathcal{S}}, f_{\mathcal{S}}, f_{k,n,\mathcal{S}})$, *then*

$$\psi_{n,\mathcal{S}} - \psi_{\mathcal{S}} = \frac{1}{n/(2-\tau)} \sum_{k=1}^{K} \left[ \sum_{i:Z_i \in \mathcal{D}_{k,a}} \phi(Z_i) - \sum_{i:Z_i \in \mathcal{D}_{k,b}} \phi_{\mathcal{S}}(Z_i) \right.$$
$$\left. + \sum_{i:Z_i \in \mathcal{D}_{k,o}} \{\phi(Z_i) - \phi_{\mathcal{S}}(Z_i)\} \right] + o_P(n^{-1/2}), \tag{3}$$

where $\phi(Z) = \dot{\mathbb{C}}(f, P; \delta_Z - P)$ and $\phi_{\mathcal{S}}(Z) = \dot{\mathbb{C}}(f_{\mathcal{S}}, P; \delta_Z - P)$. Here, $\dot{\mathbb{C}}(\tilde{f}, P; h)$ represents the Gâteaux derivative of $\tilde{P} \mapsto \mathbb{C}(\tilde{f}, \tilde{P})$ at $P$ in the direction $h$, and $\delta_z$ denotes the Dirac measure at $z$. Consequently, for any $\tau \in [0, 1)$,

$$\{n/(2 - \tau)\}^{1/2}(\psi_{n,\mathcal{S}} - \psi_{\mathcal{S}}) \xrightarrow{d} N(0, \nu_{\mathcal{S},\tau}^2)$$

as $n \to \infty$, where $\nu_{\mathcal{S},\tau}^2 = (1 - \tau)(\sigma^2 + \sigma_{\mathcal{S}}^2) + \tau\eta_{\mathcal{S}}^2$, $\sigma^2 = E\{\phi^2(Z)\}$, $\sigma_{\mathcal{S}}^2 = E\{\phi_{\mathcal{S}}^2(Z)\}$, and $\eta_{\mathcal{S}}^2 = E[\{\phi(Z) - \phi_{\mathcal{S}}(Z)\}^2]$.

*Remark* 2.2. The conditions in Theorem 2.1 are derived from Williamson et al. [5], which outline specific requirements concerning the convergence rate of estimators obtained from flexible black box prediction algorithms and the smoothness of the predictiveness measures. The validity of Theorem 2.1 relies on the asymptotic linear expansions of $\mathbb{C}(f_{k,n}, P_{k,n,I})$ and $\mathbb{C}(f_{k,n,\mathcal{S}}, P_{k,n,I})$ for $I \in \{o, a, b\}$; see Lemma B.1 or Theorem 2 in Williamson et al. [5]. Given the asymptotic linearity of $\psi_{n,\mathcal{S}}$, we can readily obtain its asymptotic distribution by observing the independence of data in $\cup_{k=1}^{K}\mathcal{D}_{k,a}$, $\cup_{k=1}^{K}\mathcal{D}_{k,b}$ and $\cup_{k=1}^{K}\mathcal{D}_{k,o}$, as well as the facts that $|\cup_{k=1}^{K}\mathcal{D}_{k,a}| = |\cup_{k=1}^{K}\mathcal{D}_{k,b}| = (1 - \tau)n/(2 - \tau)$ and $|\cup_{k=1}^{K}\mathcal{D}_{k,o}| = \tau n/(2 - \tau)$.

The asymptotic linearity of $\psi_{n,\mathcal{S}}$ in (3) exhibits distinct behaviors depending on the validity of $H_0$. Under $H_0 : \psi_{\mathcal{S}} = 0$, we have $\phi = \phi_{\mathcal{S}}$ almost surely, due to $f = f_{\mathcal{S}}$ almost surely. Consequently, the overlapping terms $\sum_{i:Z_i \in \mathcal{D}_{k,o}}\{\phi(Z_i) - \phi_{\mathcal{S}}(Z_i)\}$ for $k = 1, \ldots, K$ vanish. As a result, (3) simplifies to

$$\psi_{n,\mathcal{S}} - \psi_{\mathcal{S}} = \frac{1}{n/(2 - \tau)}\left\{\sum_{k=1}^{K}\sum_{i:Z_i \in \mathcal{D}_{k,a}}\phi(Z_i) - \sum_{k=1}^{K}\sum_{i:Z_i \in \mathcal{D}_{k,b}}\phi_{\mathcal{S}}(Z_i)\right\} + o_P(n^{-1/2}). \quad (4)$$

In this case, the dominant term compares the means of two independent samples, each with a size of $(1 - \tau)n/(2 - \tau)$. Additionally, it can be deduced that $\nu_{\mathcal{S},\tau}^2 = (1 - \tau)(\sigma^2 + \sigma_{\mathcal{S}}^2)$ under $H_0$. Conversely, under $H_1 : \psi_{\mathcal{S}} > 0$, we can reformulate (3) as

$$\psi_{n,\mathcal{S}} - \psi_{\mathcal{S}} = \frac{1}{n/(2 - \tau)}\left\{\sum_{k=1}^{K}\sum_{i:Z_i \in \mathcal{D}_{k,A}}\phi(Z_i) - \sum_{k=1}^{K}\sum_{i:Z_i \in \mathcal{D}_{k,B}}\phi_{\mathcal{S}}(Z_i)\right\} + o_P(n^{-1/2}).$$

Here, the leading term compares the means of two samples that have an overlap, with each sample having a size of $n/(2 - \tau)$.

### 2.3 Null Behaviors

To conduct the test, it is crucial to have a consistent estimator for the variance $\nu_{\mathcal{S},\tau}^2 = (1 - \tau)(\sigma^2 + \sigma_{\mathcal{S}}^2)$ under $H_0$. Following the plug-in principle, we derive an estimator $\nu_{n,\mathcal{S},\tau}^2 := (1 - \tau)K^{-1}\sum_{k=1}^{K}(\sigma_{k,n}^2 + \sigma_{k,n,\mathcal{S}}^2)$, where for each $k \in \{1, \ldots, K\}$,

$$\sigma_{k,n}^2 = \frac{1}{|\mathcal{D}_k|}\sum_{i:Z_i \in \mathcal{D}_k}\{\dot{\mathbb{C}}(f_{k,n}, P_{k,n}; \delta_{Z_i} - P_{k,n})\}^2,$$

$$\sigma_{k,n,\mathcal{S}}^2 = \frac{1}{|\mathcal{D}_k|}\sum_{i:Z_i \in \mathcal{D}_k}\{\dot{\mathbb{C}}(f_{k,n,\mathcal{S}}, P_{k,n}; \delta_{Z_i} - P_{k,n})\}^2,$$

and $P_{k,n}$ represents the empirical distribution of $\mathcal{D}_k$. The consistency of this estimator is demonstrated in Proposition 2.3.

**Proposition 2.3.** *If Conditions (C4)–(C5) in Section A hold for both tuples $(P, \mathcal{F}, f, f_{k,n})$ and $(P, \mathcal{F}_{\mathcal{S}}, f_{\mathcal{S}}, f_{k,n,\mathcal{S}})$, then, $\nu_{n,\mathcal{S},\tau}^2 \xrightarrow{P} \nu_{\mathcal{S},\tau}^2$ as $n \to \infty$ under $H_0$ for any $\tau \in [0, 1)$.*

*Remark* 2.4. Computing Gâteaux derivatives $\dot{\mathbb{C}}$ for certain predictiveness measures can be challenging. In many cases, the predictiveness measure takes the form $\mathbb{C}(\tilde{f}, \tilde{P}) = E_{\tilde{P}}\{g(Y, \tilde{f}(X))\}$, where $g$ is a known function. In such situations, it has been found that the Gâteaux derivative can be expressed as $\phi_{\tilde{f}}(z) := g(y, \tilde{f}(x)) - E\{g(Y, \tilde{f}(X))\}$; see, for example, Appendix A in Williamson et al. [5]. Consequently, when $\tilde{f} = f$, the empirical Gâteaux derivative becomes $\dot{\mathbb{C}}(f_{k,n}, P_{k,n}; \delta_z - P_{k,n}) =$

$g(y, f_{k,n}(x)) - n_k^{-1} \sum_{i: Z_i \in \mathcal{D}_k} g(Y_i, f_{k,n}(X_i))$. This formulation allows for the identification of the variance estimator $\sigma_{k,n}^2$ as the sample variance of $\{g(Y_i, f_{k,n}(X_i))\}$, simplifying the computation. Moreover, Condition (C5) is immediately satisfied (see Section B.2.1). Examples of such function $g$ include the squared loss and cross-entropy loss.

Based on Theorem 2.1 and Proposition 2.3, utilizing Slutsky's lemma, we can conclude that the normalized test statistic

$$T_\tau := \{n/(2-\tau)\}^{1/2} \psi_{n,\mathcal{S}} / \nu_{n,\mathcal{S},\tau} \xrightarrow{d} N(0,1)$$

under $H_0$ for any $\tau \in [0,1)$. For a prespecified significance level $\alpha \in (0,1)$, we reject the null hypothesis if $T_\tau > z_{1-\alpha}$, where $z_\alpha$ denotes the $\alpha$ quantile of $N(0,1)$. A summary of the entire testing procedure can be found in Section C.

*Remark* 2.5. In our asymptotic analysis of the null behavior, we explicitly exclude the case of $\tau = 1$ due to the resulting degeneracy phenomenon. Specifically, under $H_0$, when $\tau = 1$, the linear leading term of (3) becomes exactly zero. Therefore, including $\tau = 1$ can introduce a distortion in the Type I error.

### 2.4 Power Analysis

Next, we turn our attention to the power analysis of the proposed test under the alternative hypothesis $H_1 : \psi_{\mathcal{S}} > 0$.

**Theorem 2.6** (Power approximation)**.** *Suppose the conditions stated in Theorem 2.1 and Proposition 2.3 hold. Then for any $\tau \in [0,1)$, the power function $\Pr(T_\tau > z_{1-\alpha} \mid H_1) = G_{\mathcal{S},n,\alpha}(\tau) + o(1)$, where*

$$G_{\mathcal{S},n,\alpha}(\tau) = \Phi\left( -\frac{\nu_{\mathcal{S},\tau}^{(0)}}{\nu_{\mathcal{S},\tau}} z_{1-\alpha} + \frac{\{n/(2-\tau)\}^{1/2} \psi_{\mathcal{S}}}{\nu_{\mathcal{S},\tau}} \right),$$

$\nu_{\mathcal{S},\tau}^{(0)} = \{(1-\tau)(\sigma^2 + \sigma_{\mathcal{S}}^2)\}^{1/2}$ *and $\Phi$ denotes the distribution function of $N(0,1)$. Furthermore, if $Cov\{\phi(Z), \phi_{\mathcal{S}}(Z)\} \geq 0$, then $G_{\mathcal{S},n,\alpha}(\tau)$ increases with $\tau$.*

*Remark* 2.7. The form of the power function can be directly derived from Theorem 2.1 and the fact that the estimator of standard deviation $\nu_{n,\mathcal{S},\tau} \xrightarrow{p} \nu_{\mathcal{S},\tau}^{(0)}$ as $n \to \infty$ under $H_1$. Recall that $\nu_{\mathcal{S},\tau}^2 = (1-\tau)(\sigma^2 + \sigma_{\mathcal{S}}^2) + \tau\eta_{\mathcal{S}}^2$, which is provably a decreasing function of $\tau$ when $Cov\{\phi(Z), \phi_{\mathcal{S}}(Z)\} \geq 0$. Consequently, the approximate power function $G_{\mathcal{S},n,\alpha}(\tau)$ increase with $\tau$. For more details, please refer to Section B.3.

*Remark* 2.8. The requirement $Cov\{\phi(Z), \phi_{\mathcal{S}}(Z)\} \geq 0$ is relatively benign. For instance, when considering $\psi_{\mathcal{S}} = E[\{Y - E(Y \mid V)\}^2] - E[\{Y - E(Y \mid X)\}^2]$ within the framework of evaluating variable importance, this condition is readily satisfied; refer to B.3.1.

Consider the "unzipped" version of the proposed test with $\tau = 0$, which has been explored in the works of Williamson et al. [5] and Dai et al. [23] for assessing variable importance. According to Theorem 2.6, the approximate power function reduces to

$$G_{\mathcal{S},n,\alpha}(0) = \Phi\left( -z_{1-\alpha} + \frac{(n/2)^{1/2} \psi_{\mathcal{S}}}{(\sigma^2 + \sigma_{\mathcal{S}}^2)^{1/2}} \right),$$

aligning with the findings in Dai et al. [23]. In contrast, for the Zipper with $\tau \in [0,1)$, we have

$$G_{\mathcal{S},n,\alpha}(\tau) \overset{(i)}{\geq} \Phi\left( -z_{1-\alpha} + \frac{\{n/(2-\tau)\}^{1/2} \psi_{\mathcal{S}}}{(\sigma^2 + \sigma_{\mathcal{S}}^2)^{1/2}} \right) \overset{(ii)}{\geq} G_{\mathcal{S},n,\alpha}(0),$$

due to the facts that $\nu_{\mathcal{S},\tau}^{(0)} \leq \nu_{\mathcal{S},\tau}$, and $\nu_{\mathcal{S},\tau} \leq (\sigma^2 + \sigma_{\mathcal{S}}^2)^{1/2}$ if $Cov\{\phi(Z), \phi_{\mathcal{S}}(Z)\} \geq 0$. Consequently, our method surpasses the vanilla sample-splitting or cross-fitting based inferential procedures that correspond to $\tau = 0$. The improved power can be attributed to two sources: the introduction of the overlap mechanism $\tau$ (corresponding to Inequality (ii)), and the utilization of the variance estimator $\nu_{n,\mathcal{S},\tau}^2$ (Inequality (i)).

*Remark* 2.9. As discussed, $\nu_{n,\mathcal{S},\tau}^2$ is inconsistent for the limiting variance $\nu_{\mathcal{S},\tau}^2$ under $H_1$ when $0 < \tau < 1$. If the objective is to construct a valid confidence interval for the dissimilarity measure $\psi_{\mathcal{S}}$, it is crucial to use a consistent variance estimator regardless of whether $H_0$ holds or not. This can be achieved by incorporating an additional plug-in estimator of $\eta_{\mathcal{S}}^2$, which is a component of the asymptotic variance $\nu_{\mathcal{S},\tau}^2$. For detailed construction of a valid confidence interval, please refer to Section D in Appendix.

## 2.5 Efficiency-and-Degeneracy Tradeoff

Our asymptotic analysis demonstrates that the Zipper device ensures a valid testing size for any fixed slider parameter $\tau \in [0, 1)$. Moreover, as the slider $\tau$ moves away from 0, the power improves. In practical scenarios with finite sample sizes, selecting an appropriate value of $\tau$ involves a tradeoff between efficiency and degeneracy. Opting for a larger value of $\tau$ can indeed enhance the testing power. However, an excessively large $\tau$ approaching 1 would result in degeneracy and potential size inflation. This occurs because the normal approximation (4) breaks down under the null hypothesis. It is worth emphasizing that using a relatively small $\tau$ is generally safer and yields improved power compared to the vanilla splitting-based strategies with $\tau = 0$.

To achieve better power while maintaining a reliable size, we propose a simple approach for selecting $\tau$. By (4), the asymptotic normality relies on comparing means from two independent samples $\cup_{k=1}^K \mathcal{D}_{k,a}$ and $\cup_{k=1}^K \mathcal{D}_{k,b}$, each with a size of $(1 - \tau)n/(2 - \tau)$. To ensure a favorable normal approximation, we can choose the sample size $(1 - \tau)n/(2 - \tau)$ such that it meets a predetermined "large" sample size, such as $n_0 = 30$ or $50$. Say, we can specify $\tau = \tau_0 := (n - 2n_0)/(n - n_0)$. In the case of very large $n$, a truncation may be needed to safeguard against degeneracy. For example, we can set $\tau = \min\{\tau_0, 0.9\}$. Our numerical experiments show that this selection of $\tau$ achieves satisfactory performances across a wide range of scenarios. For more details on the impact of $\tau$, please refer to Section E.1.

# 3 Finite-Sample Experiments

## 3.1 Synthetic Experiments

### 3.1.1 Variable Importance Assessment

For illustration, we conduct a simulation study to evaluate the performance of the proposed Zipper method in assessing variables with zero-importance, an area of active research. We compare the empirical size and power of Zipper against several benchmark procedures. Firstly, we consider Algorithm 3 proposed in Williamson et al. [5] (referred to as WGSC-3) and the two-split test in Dai et al. [23] (DSP-Split). Both procedures involve an additional splitting of the testing data and can be seen as approximate counterparts to the proposed Zipper test with $\tau = 0$. Another benchmark procedure is Algorithm 2 from Williamson et al. [5] (WGSC-2), which can be viewed as a rough equivalent to Zipper with $\tau = 1$. Additionally, we include the data perturbation method proposed by Dai et al. [23] (DSP-Pert). For each benchmark procedure, we follow the suggestions of the respective authors to select nuisance parameters. We specify the slider parameter $\tau = \min\{\tau_0, 0.9\}$ with $n_0 = 50$ as suggested in Section 2.5.

We consider two models: one with a normal response $Y \sim N(X^\top \beta, \sigma_Y^2)$, and another with a binomial response $Y \sim \mathrm{binom}(1, \mathrm{logit}(X^\top \beta))$, where $\mathrm{logit}(t) = 1/\{1 + \exp(-t)\}$. Both models assume that $X \sim N(0, \Sigma)$, where $\Sigma = (0.2^{|i-j|})_{p \times p}$. For each model, we examine two scenarios. The first scenario is a low-dimensional setting with $p \in \{5, 10\}$ and $\beta = (\delta, \delta, 5, 0, 5, 0_{p-5})^\top$. The second scenario is a high-dimensional setting with $p \in \{200, 1000\}$ and $\beta = (\delta, \delta, 5_{0.01p}, 0_{0.99p-2}^\top)^\top$, where $a_q$ represents a $q$-dimensional vector with all entries set to $a$. In the normal model, we specify $\sigma_Y^2$ such that the signal-to-noise ratio $\beta^\top \Sigma \beta / \sigma_Y^2 = 3$ by assigning $\delta = 0$. The objective is to test whether the first two variables contribute significantly to predictions from a sequence of $n \in \{200, 500, 1000\}$ independent realizations of $(Y, X)$.

Let $\mathcal{F}$ represent a generally unrestricted model class, subjecting to a degree of sparsity under the high-dimensional settings. Consider $\mathcal{F}_{\mathcal{S}} \subseteq \mathcal{F}$ such that the prediction functions exclude the first two components of the covariates. To test the irrelevance of the first two variables in predictions, we examine $H_0 : \psi_{\mathcal{S}} = 0$ in (1). We adopt the squared loss for normal responses and the cross-entropy

Table 1: Empirical sizes (in percentage) of various testing procedures, with standard deviations in brackets.

| Model | $p$ | Zipper | WGSC-3 | DSP-Split | WGSC-2 | DSP-Pert |
|-------|-----|--------|--------|-----------|--------|----------|
| Normal | 5 | 3.9(0.19) | 5.1(0.22) | 4.6(0.21) | 0.1(0.03) | 10.2(0.30) |
|  | 1000 | 4.3(0.20) | 6.2(0.24) | 5.9(0.24) | 16.7(0.37) | 35.0(0.48) |
| Binomial | 5 | 3.7(0.19) | 3.9(0.19) | 4.2(0.20) | 0.6(0.08) | 4.0(0.20) |
|  | 1000 | 5.6(0.23) | 4.8(0.21) | 5.1(0.22) | 19.9(0.40) | 38.6(0.49) |

loss for binomial responses. The ordinary least-squares regression and the LASSO are utilized under the low-dimensional and high-dimensional scenarios, respectively. The significance level is chosen as $\alpha = 5\%$, and our experiments entail $1,000$ replications. These experiments are executed on an Intel Xeon Gold 5118 CPU @ 2.30GHz.

Table 1 displays the empirical sizes for different testing procedures with $n = 500$ and $p \in \{5, 1000\}$. The results reveal that the Zipper, WGSC-3, and DSP-Split consistently maintain the correct size across all models, as anticipated. In contrast, the WGSC-2 exhibits conservative behavior in the low-dimensional setting and inflated sizes in the high-dimensional settings, primarily due to the degeneracy phenomenon. In addition, the data perturbation method, DSP-Pert, fails to control the size in some cases, particularly in the high-dimensional settings. This instability can be attributed to the selection of the amount of perturbation.

Figure 2 depicts the empirical power of various testing methods as a function of the magnitude $\delta$ representing variable relevance, when $n = 500$ and $p \in \{5, 1000\}$. As expected, the Zipper shows a substantial improvement in power compared to the vanilla cross-fitting based approaches, WGSC-3 and DSP-Split, with WGSC-3 and DSP-Split demonstrating similar performances. Under the high-dimensional settings, the WGSC-2 and DSP-Pert exhibit higher power than Zipper, but at the expense of losing valid size control. For a comprehensive analysis of the empirical sizes and power across various combinations of $n$ and $p$, please refer to Section E.2. These additional results consistently support the conclusion that the Zipper method demonstrates reliable empirical size performance and significant power enhancement compared to that methods that utilize non-overlapping splits.

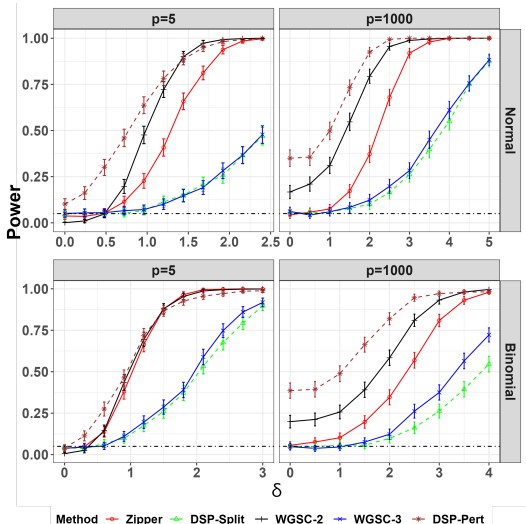

Figure 2: Empirical power of various testing methods as a function of the magnitude $\delta$ with $n = 500$ and $p \in \{5, 1000\}$. The dot-dashed horizontal line represents the intercept at $\alpha = 5\%$.

### 3.1.2 Model Specification Testing

We extend our investigation beyond the variable importance assessment problem to include an evaluation of the proposed Zipper device in addressing model specification issues. Our analysis focuses on the model defined as $Y = X\beta + \varepsilon$, where $X \sim N(0, \Sigma)$ with $\Sigma = (0.2^{|i-j|})_{p \times p}$ and $\varepsilon \sim N(0, \sigma_\varepsilon^2)$. Here, we assume that $\|\beta\|_0 = 2$, indicating the presence of two nonzero components in $\beta$, while the positions of these components remain unknown. Our objective is to test the model specification hypothesis: $H_0 : \beta = (*, *, 0_{p-2})^\top$ versus $H_1 : \|\beta\|_0 = 2$ but not $H_0$, where $*$ represents any nonzero value. To generate the data, we consider three scenarios: (i) $\beta = (0.4, 0.4, 0_{p-2})^\top$, (ii) $\beta = (0.4, 0, 0.4, 0_{p-3})^\top$, and (iii) $\beta = (0, 0, 0.4, 0.4, 0_{p-4})^\top$. We determine the value of $\sigma_\varepsilon^2$ such that the signal-to-noise ratio $\beta^\top \Sigma \beta / \sigma_\varepsilon^2 = 1$. Subsequently, we generate independent realizations $(Y_i, X_i)$ for $i = 1, \ldots, n$, with $n = 500$ and $p \in \{5, 1000\}$. To estimate $\beta$, we employ ordinary least squares under $H_0$, and perform the best

two subset selection under $H_1$. For the case when $p = 5$, we conduct an exhaust search. For the case when $p = 1000$, we utilize the abess algorithm [31] to approximate the solutions. The results are summarized in Table 2, where we observe that under Scenario (i), corresponding to $H_0$, Zipper maintains correct size control. Furthermore, under Scenarios (ii) and (iii), corresponding to $H_1$, Zipper exhibits substantial improvements in power when compared to the vanilla cross-fitting based approaches, WGSC-3 and DSP-Split.

Table 2: Empirical sizes and powers (in percentage) for the model specification test, with standard deviations in brackets.

| $p$ | 5 | | | | 1000 | | | |
|---|---|---|---|---|---|---|---|---|
| Scenerio | Zipper | WGSC-3 | DSP-Split | WGSC-2 | Zipper | WGSC-3 | DSP-Split | WGSC-2 |
| (i) | 4.3(0.20) | 6.2(0.22) | 5.6(0.20) | 0.0(0.00) | 4.2(0.19) | 5.5(0.20) | 6.5(0.21) | 16.6(0.36) |
| (ii) | 96.9(0.17) | 31.2(0.46) | 34.9(0.46) | 100.0(0.00) | 94.2(0.22) | 29.8(0.46) | 31.4(0.46) | 97.3(0.16) |
| (iii) | 100.0(0.00) | 81.4(0.39) | 79.3(0.38) | 100.0(0.00) | 100.0(0.00) | 81.3(0.40) | 78.1(0.41) | 100.0(0.00) |

## 3.2 Real-Data Examples

### 3.2.1 MNIST Handwritten Dataset

We apply the Zipper method to the widely used MNIST handwritten digit dataset [32]. The MNIST dataset consists of size-normalized and center-aligned handwritten digit images, each represented as a $28 \times 28$ pixel grid (resulting in $p = 28^2 = 784$). For our analysis, we specifically extract subsets of the dataset representing the digits 7 and 9, following Dai et al. [23], resulting in a total of $n = 14251$ images.

In Figure 3, we calculate and graphically represent the average grayscale pixel values for images sharing the same numerals. We divide each image into nine distinct regions, as shown by the blank squares in Figure 3, with the objective of detecting regions that can effectively distinguish between these two digits. To achieve this, we perform a sequence of variable importance testing to assess the relevance of each region in making predictions while considering the remaining regions. Given the nature of the data, we employ a Convolutional Neural Network (CNN) as the underlying model, leveraging its proven effectiveness in image analysis. In the Zipper approach, we select the slider parameter $\tau$ such that $n_0 = 50$, as recommended in the manuscript. As a benchmark, we adopt WGSC-3 [5] (equivalent to DSP-Split [23]), which produces valid size, aligning with our approach. We set the predefined significance level for each test as $\alpha = 0.05/9$, applying the Bonferroni correction to account for multiple comparisons. The discovered regions, highlighted in red, are presented in Figure 3. Our findings indicate that the Zipper method outperforms WGSC-3 in identifying critical regions with greater efficacy.

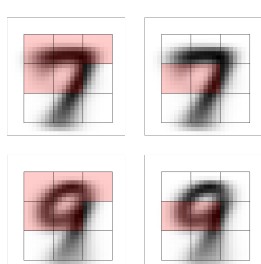

Figure 3: Hypothesis regions (blank squares) and important discoveries (squares filled in red) comparing the Zipper method (left column) with WGSC-3 (right column).

### 3.2.2 Bodyfat Dataset

We expand the application of our Zipper method to the bodyfat dataset [33], which provides an estimate of body fat percentages obtained through underwater weighing, along with various body circumference measurements from a sample of $n = 252$ men. Our objective is to conduct marginal variable importance tests for each body circumference while considering potential influences from essential attributes such as age, weight, and height. To accurately estimate the relevant regression functions within this dataset, we employ the random forest as our modeling technique. Table 3 presents the resulting p-values obtained from the Zipper and WGSC-3 methods. By applying the Bonferroni correction, the Zipper method identifies both Abdomen and Hip as significant factors at the significance level of $\alpha = 0.05/10$. In contrast, WGSC-3 suggests only Abdomen as important. It is worth noting that a recent study by Zhu et al. [34] proposed the formula (Waist+Hip)/Height as a straightforward evaluation index for body fat. Remarkably, our finds align with this fact, further supporting the validity and relevance of our Zipper method in identifying key factors.

Table 3: P-values obtained from the Zipper and WGSC-3 methods for each marginal test regarding the relevance of the body circumference.

| Body Part | Neck | Chest | Abdomen | Hip | Thigh | Knee | Ankle | Biceps | Forearm | Wrist |
|-----------|------|-------|---------|-----|-------|------|-------|--------|---------|-------|
| Zipper | 0.98 | 0.10 | $5.48 \times 10^{-10}$ | $4.01 \times 10^{-4}$ | 0.10 | 0.03 | 0.20 | 0.26 | 0.35 | 0.02 |
| WGSC-3 | 0.12 | 0.01 | $9.30 \times 10^{-4}$ | 0.29 | 0.01 | 0.06 | 0.36 | 0.18 | 0.69 | 0.05 |

## 4 Concluding Remarks

In this paper, we introduce Zipper, a simple yet effective device designed to address the issue of degeneracy in algorithm/model-agnostic inference. The mechanism of Zipper involves the recycling of data usage by constructing two overlapping data splits within the testing samples, which holds potential for independent exploration. A key component of Zipper is the slider parameter, which introduces an efficiency-and-degeneracy tradeoff. To ensure reliable inference, we propose a simple selection criterion by ensuring a large sample size to render asymptotic normality under the null hypothesis. Other data-adaptive strategies are possible and merit further investigation. Moreover, the predictiveness-comparison-based framework allows for the utilization of alternative forms of two-sample tests, such as rank-based methods. This capability proves beneficial when dealing with data exhibiting heavy-tailed distributions or outliers. Furthermore, incorporating the Zipper device into *large-scale comparisons* to achieve error rate control warrants additional research. We can conduct a sequence of variable importance tests, each aimed at assessing the relevance of a specific variable $X_j$ in the predictive model while controlling for a global error rate. This procedure necessitates the fitting of $p + 1$ models: one that includes all variables and $p$ null models, each excluding a distinct variable. Such a process is computationally demanding. Moreover, accurately controlling error rates presents a considerable challenge due to complex dependency structures among the p-values.

## Acknowledgements

We would like to acknowledge the anonymous area chair and reviewers for their valuable comments and suggestions. Zou was supported by the National Key R&D Program of China (Grant Nos. 2022YFA1003703, 2022YFA1003800) and the National Natural Science Foundation of China (Grant Nos. 11925106, 12231011, 11931001, 12226007, 12326325). Wang was supported by the National Natural Science Foundation of China (Grant No. 12471255) and the Natural Science Foundation of Shanghai (Grant No. 23ZR1419400).

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

# Appendix

## A  Conditions

Assume $P \in \mathcal{M}$ for a rich class $\mathcal{M}$ of distributions. Define the linear space $\mathcal{H} = \{c(\tilde{P}_1 - \tilde{P}_2) : c \geq 0, \tilde{P}_1, \tilde{P}_2 \in \mathcal{M}\}$. For each $h \in \mathcal{H}$, let $\|h\|_\infty = \sup_z |\tilde{F}_1(z) - \tilde{F}_2(z)|$, where $\tilde{F}_j$ denotes the distribution function corresponding to $\tilde{P}_j$ for $j = 1, 2$. Consider $\mathcal{F}$ as the class of predictions functions endowed with a norm $\|\cdot\|_\mathcal{F}$. Let $f \in \arg\max_{\tilde{f} \in \mathcal{F}} \mathbb{C}(\tilde{f}, P)$, and $\{f_{k,n}\}_{k=1}^K$ be a sequence of estimators of $f$. For each $1 \leq k \leq K$, define $a_{k,n} : z \mapsto \dot{\mathbb{C}}(f_{k,n}, P; \delta_z - P) - \dot{\mathbb{C}}(f, P; \delta_z - P)$ and $b_{k,n} : z \mapsto \dot{\mathbb{C}}(f_{k,n}, P_{k,n}; \delta_z - P_{k,n}) - \dot{\mathbb{C}}(f_{k,n}, P; \delta_z - P)$. The following conditions are imposed on the tuple $(P, \mathcal{F}, f, f_{k,n})$.

(C1) There exists some constant $C > 0$ such that, for each sequence $\tilde{f}_1, \tilde{f}_2, \ldots \in \mathcal{F}$ such that $\|\tilde{f}_j - f\|_\mathcal{F} \to 0$, $|\mathbb{C}(\tilde{f}_j, P) - \mathbb{C}(f, P)| \leq C\|\tilde{f}_j - f\|_\mathcal{F}^2$ for each $j$ large enough;

(C2) There exists some constant $\delta > 0$ such that for each sequence $\epsilon_1, \epsilon_2, \ldots \in \mathbb{R}$ and $h, h_1, h_2, \ldots \in \mathcal{H}$ satisfying that $\epsilon_j \to 0$ and $\|h_j - h\|_\infty \to 0$, it holds that

$$\sup_{\tilde{f} \in \mathcal{F} : \|\tilde{f} - f\|_\mathcal{F} < \delta} \left| \frac{\mathbb{C}(\tilde{f}, P + \epsilon_j h_j) - \mathbb{C}(\tilde{f}, P)}{\epsilon_j} - \dot{\mathbb{C}}(\tilde{f}, P; h_j) \right| \to 0;$$

(C3) For each $1 \leq k \leq K$, $\|f_{k,n} - f\|_\mathcal{F} = o_P(n^{-1/4})$;

(C4) For each $1 \leq k \leq K$, $\int a_{k,n}^2 dP = o_P(1)$;

(C5) For each $1 \leq k \leq K$, $\int b_{k,n}^2 dP = o_P(1)$.

Condition (C1) pertains to the optimality of the prediction function $f$, eliminating first-order estimation biases. Condition (C2) requires Hadamard differentiability of the predictiveness criterion. Condition (C3) demands accurate estimation of $f$, rendering second-order terms negligible. Condition (C4) controls the remainder terms. Condition (C5) ensures consistent variance estimators.

## B  Proofs

### B.1  Proof of Theorem 2.1

**Lemma B.1.** *If Conditions (C1)–(C4) specified in Appendix hold for both tuples $(P, \mathcal{F}, f, f_{k,n})$ and $(P, \mathcal{F}_\mathcal{S}, f_\mathcal{S}, f_{k,n,\mathcal{S}})$, then, for each $I \in \{o, a, b\}$ and $k \in \{1, \ldots, K\}$,*

$$\mathbb{C}(f_{k,n}, P_{k,n,I}) - \mathbb{C}(f, P) = \frac{1}{|\mathcal{D}_{k,I}|} \sum_{i : Z_i \in \mathcal{D}_{k,I}} \phi(Z_i) + o_P(|\mathcal{D}_{k,I}|^{-1/2}) \text{ and}$$

$$\mathbb{C}(f_{k,n,\mathcal{S}}, P_{k,n,I}) - \mathbb{C}(f_\mathcal{S}, P) = \frac{1}{|\mathcal{D}_{k,I}|} \sum_{i : Z_i \in \mathcal{D}_{k,I}} \phi_\mathcal{S}(Z_i) + o_P(|\mathcal{D}_{k,I}|^{-1/2}).$$

*Proof.* See Theorem 2 in Williamson et al. [5]. ∎

Denote $m_k = |\mathcal{D}_{k,A}| = |\mathcal{D}_{k,B}|$. Since $|\mathcal{D}_{k,a}| = |\mathcal{D}_{k,b}| = (1 - \tau)m_k$ and $|\mathcal{D}_{k,o}| = \tau m_k$, we have $m_k = n/\{(2 - \tau)K\}$ due to $|\mathcal{D}_{k,a}| + |\mathcal{D}_{k,b}| + |\mathcal{D}_{k,o}| = n/K$. By Lemma B.1, we have

$$\mathbb{C}_{k,n} - \mathbb{C}(f, P) = \tau\{\mathbb{C}(f_{k,n}, P_{k,n,o}) - \mathbb{C}(f, P)\} + (1 - \tau)\{\mathbb{C}(f_{k,n}, P_{k,n,a}) - \mathbb{C}(f, P)\}$$

$$= \frac{1}{m_k} \sum_{i : Z_i \in \mathcal{D}_{k,A}} \phi(Z_i) + o_P(m_k^{-1/2}).$$

Since the number of folds $K$ is fixed, we have

$$\frac{1}{K}\sum_{k=1}^{K}\mathbb{C}_{k,n} - \mathbb{C}(f,P) = \frac{1}{K}\sum_{k=1}^{K}\left\{\frac{1}{m_k}\sum_{i:Z_i\in\mathcal{D}_{k,A}}\phi(Z_i) + o_P(m_k^{-1/2})\right\}$$

$$= \frac{1}{K}\sum_{k=1}^{K}\frac{1}{m_k}\sum_{i:Z_i\in\mathcal{D}_{k,A}}\phi(Z_i) + o_P(n^{-1/2}). \tag{5}$$

Similarly, we conclude that

$$\frac{1}{K}\sum_{k=1}^{K}\mathbb{C}_{k,n,\mathcal{S}} - \mathbb{C}(f_\mathcal{S},P) = \frac{1}{K}\sum_{k=1}^{K}\frac{1}{m_k}\sum_{i:Z_i\in\mathcal{D}_{k,B}}\phi_\mathcal{S}(Z_i) + o_P(n^{-1/2}). \tag{6}$$

Combining (5) and (6), we obtain

$$\psi_{n,\mathcal{S}} - \psi_\mathcal{S}$$

$$= \frac{1}{n/(2-\tau)}\sum_{k=1}^{K}\left[\sum_{i:Z_i\in\mathcal{D}_{k,a}}\phi(Z_i) - \sum_{i:Z_i\in\mathcal{D}_{k,b}}\phi_\mathcal{S}(Z_i) + \sum_{i:Z_i\in\mathcal{D}_{k,o}}\{\phi(Z_i)-\phi_\mathcal{S}(Z_i)\}\right] + o_P(n^{-1/2}).$$

By applying the standard central limit theorem,

$$\frac{\{(1-\tau)n/(2-\tau)\}^{1/2}}{K}\sum_{k=1}^{K}\frac{1}{(1-\tau)m_k}\sum_{i:Z_i\in\mathcal{D}_{k,a}}\phi(Z_i) \xrightarrow{d} N(0,\sigma^2),\text{ and}$$

$$\frac{\{(1-\tau)n/(2-\tau)\}^{1/2}}{K}\sum_{k=1}^{K}\frac{1}{(1-\tau)m_k}\sum_{i:Z_i\in\mathcal{D}_{k,b}}\phi_\mathcal{S}(Z_i) \xrightarrow{d} N(0,\sigma_\mathcal{S}^2),$$

as $n\to\infty$. If $\psi_\mathcal{S} > 0$, we have

$$\frac{\{\tau n/(2-\tau)\}^{1/2}}{K}\sum_{k=1}^{K}\frac{1}{\tau m_k}\sum_{i:Z_i\in\mathcal{D}_{k,o}}\{\phi(Z_i)-\phi_\mathcal{S}(Z_i)\} \xrightarrow{d} N(0,\eta_\mathcal{S}^2).$$

Hence,

$$\{n/(2-\tau)\}^{1/2}(\psi_{n,\mathcal{S}}-\psi_\mathcal{S}) \xrightarrow{d} N(0,\nu_{\mathcal{S},\tau}^2),$$

where $\nu_{\mathcal{S},\tau}^2 = (1-\tau)(\sigma^2+\sigma_\mathcal{S}^2) + \tau\eta_\mathcal{S}^2$.

## B.2  Proof of Proposition 2.3

It suffices to show that $\sigma_{k,n}^2 \xrightarrow{p} \sigma^2$ as $n\to\infty$. Since

$$\dot{\mathbb{C}}(f_{k,n},P_{k,n};\delta_z - P_{k,n}) = \dot{\mathbb{C}}(f_{k,n},P_{k,n};\delta_z-P_{k,n}) - \dot{\mathbb{C}}(f_{k,n},P;\delta_z-P)$$
$$+ \dot{\mathbb{C}}(f_{k,n},P;\delta_z-P) - \dot{\mathbb{C}}(f,P;\delta_z-P) + \dot{\mathbb{C}}(f,P;\delta_z-P),$$

we have

$$\sigma^2_{k,n}$$

$$= \frac{1}{n_k}\Bigg[ \sum_{i:Z_i \in \mathcal{D}_k} \{\dot{\mathbb{C}}(f_{k,n}, P_{k,n}; \delta_{Z_i} - P_{k,n}) - \dot{\mathbb{C}}(f_{k,n}, P; \delta_{Z_i} - P)\}^2$$

$$+ \sum_{i:Z_i \in \mathcal{D}_k} \{\dot{\mathbb{C}}(f_{k,n}, P; \delta_{Z_i} - P) - \dot{\mathbb{C}}(f, P; \delta_{Z_i} - P)\}^2$$

$$+ \sum_{i:Z_i \in \mathcal{D}_k} \{\dot{\mathbb{C}}(f, P; \delta_{Z_i} - P)\}^2$$

$$+ 2\sum_{i:Z_i \in \mathcal{D}_k} \{\dot{\mathbb{C}}(f_{k,n}, P_{k,n}; \delta_{Z_i} - P_{k,n}) - \dot{\mathbb{C}}(f_{k,n}, P; \delta_{Z_i} - P)\}\dot{\mathbb{C}}(f, P; \delta_{Z_i} - P) \quad (7)$$

$$+ 2\sum_{i:Z_i \in \mathcal{D}_k} \{\dot{\mathbb{C}}(f_{k,n}, P; \delta_{Z_i} - P) - \dot{\mathbb{C}}(f, P; \delta_{Z_i} - P)\}\dot{\mathbb{C}}(f, P; \delta_{Z_i} - P)$$

$$+ 2\sum_{i:Z_i \in \mathcal{D}_k} \{\dot{\mathbb{C}}(f_{k,n}, P_{k,n}; \delta_{Z_i} - P_{k,n}) - \dot{\mathbb{C}}(f_{k,n}, P; \delta_{Z_i} - P)\}$$

$$\times \{\dot{\mathbb{C}}(f_{k,n}, P; \delta_{Z_i} - P) - \dot{\mathbb{C}}(f, P; \delta_{Z_i} - P)\}\Bigg].$$

For any $\epsilon > 0$, by Markov's inequality and Condition (C5),

$$\Pr\Bigg[\frac{1}{n_k} \sum_{i:Z_i \in \mathcal{D}_k} \{\dot{\mathbb{C}}(f_{k,n}, P_{k,n}; \delta_{Z_i} - P_{k,n}) - \dot{\mathbb{C}}(f_{k,n}, P; \delta_{Z_i} - P)\}^2 \geq \epsilon \mid \mathcal{D}_{-k}\Bigg]$$

$$\leq \frac{1}{\epsilon}E[\{\dot{\mathbb{C}}(f_{k,n}, P_{k,n}; \delta_{Z_i} - P_{k,n}) - \dot{\mathbb{C}}(f_{k,n}, P; \delta_{Z_i} - P)\}^2 \mid \mathcal{D}_{-k}]$$

$$= \frac{1}{\epsilon}\int \{b_{k,n}(z)\}^2 dP(z) \xrightarrow{p} 0.$$

Then, by the law of total expectation, the first term of (7) is $o_P(1)$. Similarly, the second term is $o_P(1)$. Specifically, by Condition (C4), for any $\epsilon > 0$,

$$\Pr\Bigg[\frac{1}{n_k} \sum_{i:Z_i \in \mathcal{D}_k} \{\dot{\mathbb{C}}(f_{k,n}, P; \delta_{Z_i} - P) - \dot{\mathbb{C}}(f, P; \delta_{Z_i} - P)\}^2 \geq \epsilon \mid \mathcal{D}_{-k}\Bigg]$$

$$\leq \frac{1}{\epsilon}E[\{\dot{\mathbb{C}}(f_{k,n}, P; \delta_Z - P) - \dot{\mathbb{C}}(f, P; \delta_Z - P)\}^2 \mid \mathcal{D}_{-k}]$$

$$= \frac{1}{\epsilon}\int \{a_{k,n}(z)\}^2 dP(z) \xrightarrow{p} 0.$$

For the third term, by the law of large numbers,

$$\frac{1}{n_k} \sum_{i:Z_i \in \mathcal{D}_k} \dot{\mathbb{C}}(f, P; \delta_{Z_i} - P)^2 \xrightarrow{p} E\{\dot{\mathbb{C}}(f, P; \delta_Z - P)^2\} = \sigma^2,$$

as $n \to \infty$. For the fourth term, we have

$$\left|\frac{1}{n_k} \sum_{i:Z_i \in \mathcal{D}_k} \{\dot{\mathbb{C}}(f_{k,n}, P_{k,n}; \delta_{Z_i} - P_{k,n}) - \dot{\mathbb{C}}(f_{k,n}, P; \delta_{Z_i} - P)\}\dot{\mathbb{C}}(f, P; \delta_{Z_i} - P)\right|$$

$$\leq \left[\frac{1}{n_k} \sum_{i:Z_i \in \mathcal{D}_k} \{\dot{\mathbb{C}}(f_{k,n}, P_{k,n}; \delta_{Z_i} - P_{k,n}) - \dot{\mathbb{C}}(f_{k,n}, P; \delta_{Z_i} - P)\}^2\right]^{1/2}$$

$$\times \left[\frac{1}{n_k} \sum_{i:Z_i \in \mathcal{D}_k} \dot{\mathbb{C}}(f, P; \delta_{Z_i} - P)^2\right]^{1/2} = o_P(1).$$

Similarly, both of the last two terms are $o_P(1)$. Hence, it follows that $\sigma^2_{k,n} \xrightarrow{p} \sigma^2$.

By similar arguments, $\sigma^2_{k,n,\mathcal{S}} \xrightarrow{p} \sigma^2_{\mathcal{S}}$ and thus $\nu^2_{n,\mathcal{S},\tau} \xrightarrow{p} \nu_{\mathcal{S},\tau}$.

### B.2.1 Verification of Condition (C5) in Remark 2.4

Suppose $\mathbb{C}(f, P) = E_P[g\{Y, f(X)\}]$. Then

$$\dot{\mathbb{C}}(f_{k,n}, P_{k,n}; \delta_z - P_{k,n}) = g\{y, f_{k,n}(x)\} - \frac{1}{n_k} \sum_{i:Z_i \in \mathcal{D}_k} g\{Y_i, f_{k,n}(X_i)\},$$

$$\dot{\mathbb{C}}(f_{k,n}, P; \delta_z - P) = g\{y, f_{k,n}(x)\} - E_P[g\{Y, f_{k,n}(X)\}].$$

By noting that

$$E[\{\dot{\mathbb{C}}(f_{k,n}, P_{k,n}; \delta_z - P_{k,n}) - \dot{\mathbb{C}}(f_{k,n}, P; \delta_z - P)\}^2 \mid \mathcal{D}_{-k}]$$

$$= E\left(\left[\frac{1}{n_k} \sum_{i:Z_i \in \mathcal{D}_k} g\{Y_i, f_{k,n}(X_i)\}\right]^2 \mid \mathcal{D}_{-k}\right) - E[g\{Y, f_{k,n}(X)\} \mid \mathcal{D}_{-k}]^2$$

$$= \frac{1}{n_k} Var[g\{Y, f_{k,n}(X)\} \mid \mathcal{D}_{-k}] \xrightarrow{p} 0,$$

the conclusion follows.

### B.3 Proof of Theorem 2.6

By Theorem 2.1, for any $\tau \in [0, 1)$,

$$\{n/(2 - \tau)\}^{1/2}(\psi_{n,\mathcal{S}} - \psi_{\mathcal{S}}) \xrightarrow{d} N(0, \nu_{\mathcal{S},\tau}^2)$$

as $n \to \infty$, where $\nu_{\mathcal{S},\tau}^2 = (1 - \tau)(\sigma^2 + \sigma_{\mathcal{S}}^2) + \tau\eta_{\mathcal{S}}^2$, $\sigma^2 = E\{\phi^2(Z)\}$, $\sigma_{\mathcal{S}}^2 = E\{\phi_{\mathcal{S}}^2(Z)\}$, and $\eta_{\mathcal{S}}^2 = E[\{\phi(Z) - \phi_{\mathcal{S}}(Z)\}^2]$. By examining the proof of Proposition 2.3, we conclude that $\nu_{n,\mathcal{S},\tau} \xrightarrow{p} \nu_{\mathcal{S},\tau}^{(0)}$ under $H_1$. Consequently, the power function is

$$\Pr(T_\tau > z_{1-\alpha} \mid H_1)$$

$$= \Pr\left(\frac{\nu_{\mathcal{S},\tau}^{(0)}}{\nu_{n,\mathcal{S},\tau}} \frac{\{n/(2 - \tau)\}^{1/2}(\psi_{n,\mathcal{S}} - \psi_{\mathcal{S}})}{\nu_{\mathcal{S},\tau}} > \frac{\nu_{\mathcal{S},\tau}^{(0)}}{\nu_{\mathcal{S},\tau}} z_{1-\alpha} - \frac{\{n/(2 - \tau)\}^{1/2}\nu_{\mathcal{S},\tau}^{(0)}\psi_{\mathcal{S}}}{\nu_{n,\mathcal{S},\tau}\nu_{\mathcal{S},\tau}} \mid H_1\right)$$

$$= \Phi\left(-\frac{\nu_{\mathcal{S},\tau}^{(0)}}{\nu_{\mathcal{S},\tau}} z_{1-\alpha} + \frac{\{n/(2 - \tau)\}^{1/2}\psi_{\mathcal{S}}}{\nu_{\mathcal{S},\tau}}\right) + o(1)$$

$$= G_{\mathcal{S},n,\alpha}(\tau) + o(1).$$

Given that $\eta_{\mathcal{S}} = \sigma^2 + \sigma_{\mathcal{S}}^2 - 2Cov\{\phi(Z), \phi_{\mathcal{S}}(Z)\}$, the asymptotic variance $\nu_{\mathcal{S},\tau}^2$ can be expressed as $\sigma^2 + \sigma_{\mathcal{S}}^2 - 2\tau Cov\{\phi(Z), \phi_{\mathcal{S}}(Z)\}$. Let

$$\varphi_1(\tau) = \frac{(1 - \tau)(\sigma^2 + \sigma_{\mathcal{S}}^2)}{\sigma^2 + \sigma_{\mathcal{S}}^2 - 2\tau Cov\{\phi(Z), \phi_{\mathcal{S}}(Z)\}} \quad \text{and}$$

$$\varphi_2(\tau) = \frac{\{n/(2 - \tau)\}^{1/2}\psi_{\mathcal{S}}}{(\sigma^2 + \sigma_{\mathcal{S}}^2 - 2\tau Cov\{\phi(Z), \phi_{\mathcal{S}}(Z)\})^{1/2}}.$$

Since $2Cov\{\phi(Z), \phi_{\mathcal{S}}(Z)\} \leq \sigma^2 + \sigma_{\mathcal{S}}^2$ and $\tau \in [0, 1)$, it follows that $\varphi_1(\tau) > 0$ and $\varphi_2(\tau) > 0$. When $Cov\{\phi(Z), \phi_{\mathcal{S}}(Z)\} \geq 0$, we observe that $\varphi_1(\tau)$ is monotonically decreasing, and $\varphi_2(\tau)$ is monotonically increasing with respect to $\tau$. Therefore, the approximate power function $G_{\mathcal{S},n,\alpha}(\tau)$ increase with $\tau$.

### B.3.1 On Remark 2.8

Here we demonstrate the validity of the condition $Cov\{\phi(Z), \phi_{\mathcal{S}}(Z)\} \geq 0$ in assessing variable importance when utilizing the square loss. Consider the scenario where $Y = E(Y \mid X) + \varepsilon$, with the noise $\varepsilon$ assumed to be independent of the covariate $X$. We seek to evaluate the significance of a specific set of covariates $U$ in predicting the response, where $X = (U^\top, V^\top)^\top$. Assuming

$E(Y \mid X) \in \mathcal{F}$ and $E(Y \mid V) \in \mathcal{F}_\mathcal{S}$, the optimal prediction functions within $\mathcal{F}$ and $\mathcal{F}_\mathcal{S}$ are $f(X) = E(Y \mid X)$ and $f_\mathcal{S}(V) = E(Y \mid V)$, respectively.

Referring to Remark 2.4, the covariance $Cov\{\phi(Z), \phi_\mathcal{S}(Z)\}$ can be expressed as

$$Cov\{\phi(Z), \phi_\mathcal{S}(Z)\} = Cov[\{Y - f(X)\}^2, \{Y - f_\mathcal{S}(V)\}^2]$$
$$= Var(\varepsilon^2) + Cov[\varepsilon^2, \{f(X) - f_\mathcal{S}(V)\}^2] + 2Cov[\varepsilon^2, \varepsilon\{f(X) - f_\mathcal{S}(V)\}].$$

It's evident that the first two terms are non-negative. Regarding the third term, since $\varepsilon$ is independent of $X$ and $Ef(X) \mid V = f_\mathcal{S}(V)$, we deduce

$$Cov[\varepsilon^2, \varepsilon\{f(X) - f_\mathcal{S}(V)\}] = E[\varepsilon^3\{f(X) - f_\mathcal{S}(V)\}] - E(\varepsilon^2)E[\varepsilon\{f(X) - f_\mathcal{S}(V)\}]$$
$$= E(\varepsilon^3)E\{f(X) - f_\mathcal{S}(V)\} - E(\varepsilon^2)E(\varepsilon)E\{f(X) - f_\mathcal{S}(V)\}$$
$$= E(\varepsilon^3)E[E\{f(X) - f_\mathcal{S}(V) \mid V\}]$$
$$= 0.$$

Thus, $Cov\{\phi(Z), \phi_\mathcal{S}(Z)\} \geq 0$ follows.

## C  The Zipper Algorithm

---

**Algorithm 1** The algorithm for the proposed Zipper testing procedure

---

**Input:** Observed data $\{Z_i\}_{i=1}^n$, number of folds $K$, and slider parameter $\tau \in [0, 1)$
Randomly partition $\{Z_i\}_{i=1}^n$ into $K$ disjoint folds $\mathcal{D}_1, \ldots, \mathcal{D}_K$
**for** $k = 1, \ldots, K$ **do**
    Using data in $\{\mathcal{D}_j\}_{j=1}^K \setminus \mathcal{D}_k$, construct estimators $f_{k,n}$ of $f$ and $f_{k,n,\mathcal{S}}$ of $f_\mathcal{S}$
    Using data in $\mathcal{D}_k$, construct estimator $P_{k,n}$ of $P$
    Randomly divide $\mathcal{D}_k$ into two splits $\mathcal{D}_{k,A}$ and $\mathcal{D}_{k,B}$ of roughly equal size $m_k$, with an overlap $\mathcal{D}_{k,o}$ such that $|\mathcal{D}_{k,o}| = \lfloor \tau m_k \rfloor$. Let $\mathcal{D}_{k,a} = \mathcal{D}_{k,A} \setminus \mathcal{D}_{k,o}$ and $\mathcal{D}_{k,b} = \mathcal{D}_{k,B} \setminus \mathcal{D}_{k,o}$. Using data in $\mathcal{D}_{k,I}$, construct estimators $P_{k,n,I}$ for $I \in \{o, a, b\}$
    Compute $\mathbb{C}_{k,n} = \tau\mathbb{C}(f_{k,n}, P_{k,n,o}) + (1 - \tau)\mathbb{C}(f_{k,n}, P_{k,n,a})$ and $\sigma_{k,n}^2 = n_k^{-1}\sum_{i:Z_i \in \mathcal{D}_k}\{\dot{\mathbb{C}}(f_{k,n}, P_{k,n}; \delta_{Z_i} - P_{k,n})\}^2$, where $n_k = |\mathcal{D}_k|$
    Compute $\mathbb{C}_{k,n,\mathcal{S}} = \tau\mathbb{C}(f_{k,n,\mathcal{S}}, P_{k,n,o}) + (1 - \tau)\mathbb{C}(f_{k,n,\mathcal{S}}, P_{k,n,b})$ and $\sigma_{k,n,\mathcal{S}}^2 = n_k^{-1}\sum_{i:Z_i \in \mathcal{D}_k}\{\dot{\mathbb{C}}(f_{k,n,\mathcal{S}}, P_{k,n}; \delta_{Z_i} - P_{k,n})\}^2$
**end for**
Compute $\mathbb{C}_n = K^{-1}\sum_{k=1}^K \mathbb{C}_{k,n}$, $\mathbb{C}_{n,\mathcal{S}} = K^{-1}\sum_{k=1}^K \mathbb{C}_{k,n,\mathcal{S}}$ and estimator $\psi_{n,\mathcal{S}} = \mathbb{C}_n - \mathbb{C}_{n,\mathcal{S}}$ of $\psi_\mathcal{S}$
Compute $\nu_{n,\mathcal{S},\tau}^2 = (1 - \tau)K^{-1}\sum_{k=1}^K (\sigma_{k,n}^2 + \sigma_{k,n,\mathcal{S}}^2)$
**Output:** P-value $1 - \Phi(\{n/(2 - \tau)\}^{1/2}\psi_{n,\mathcal{S}}/\nu_{n,\mathcal{S},\tau})$, where $\Phi$ denotes the distribution function of $N(0, 1)$

---

## D  The Confidence Interval for the Dissimilarity Measure

To construct a valid confidence interval for the dissimilarity measure $\psi_\mathcal{S}$, it is crucial to use a consistent variance estimator irrespective of whether $H_0$ holds or not. This can be achieved by incorporating an additional plug-in estimator of $\eta_\mathcal{S}^2$, which is a component of the asymptotic variance $\nu_{\mathcal{S},\tau}^2$. Let $\nu_{n,\mathcal{S},\tau}^2 = K^{-1}\sum_{k=1}^K\{(1 - \tau)(\sigma_{k,n}^2 + \sigma_{k,n,\mathcal{S}}^2) + \tau\eta_{k,n,\mathcal{S}}^2\}$, where for each $k \in \{1, \ldots, K\}$,

$$\eta_{k,n,\mathcal{S}}^2 = \frac{1}{n_k}\sum_{i:Z_i \in \mathcal{D}_k}\{\dot{\mathbb{C}}(f_{k,n}, P_{k,n}; \delta_{Z_i} - P_{k,n}) - \dot{\mathbb{C}}(f_{k,n,\mathcal{S}}, P_{k,n}; \delta_{Z_i} - P_{k,n})\}^2.$$

The confidence interval for $\psi_\mathcal{S}$ can be formulated as

$$\left(\psi_{n,\mathcal{S}} - \frac{z_{\alpha/2}\nu_{n,\mathcal{S},\tau}}{\{n/(2 - \tau)\}^{1/2}}, \psi_{n,\mathcal{S}} + \frac{z_{\alpha/2}\nu_{n,\mathcal{S},\tau}}{\{n/(2 - \tau)\}^{1/2}}\right),$$

which has an asymptotic coverage of $1 - \alpha$.

Table 4: Empirical sizes (in percentage) of the Zipper method against different values of $\tau$ with $n = 500$.

| Model | $p$ | $\tau$ | | | | | | | |
|---|---|---|---|---|---|---|---|---|---|
| | | 0 | 0.2 | 0.4 | 0.6 | 0.8 | 0.9 | 0.95 | 0.99 |
| Normal | 5 | 4.6 | 3.2 | 3.2 | 4.4 | 4.9 | 4.5 | 3.7 | 2.2 |
| | 1000 | 5.9 | 5.9 | 5.8 | 4.2 | 5.6 | 7.4 | 8.2 | 11.1 |
| Binomial | 5 | 4.2 | 3.3 | 2.4 | 2.2 | 3.6 | 3.1 | 3.0 | 3.5 |
| | 1000 | 5.1 | 5.7 | 4.2 | 5.1 | 5.6 | 9.5 | 11.7 | 20.9 |

Table 5: Empirical power (in percentage) of the Zipper method against different values of $\tau$ with $n = 500$. For the normal model, we set $\delta = 2$ for the low-dimensional setting and $\delta = 5$ for the high-dimensional setting. For the Binomial model, we set $\delta = 3$ for the low-dimensional setting and $\delta = 5$ for the high-dimensional setting.

| Model | $p$ | $\tau$ | | | | | |
|---|---|---|---|---|---|---|---|
| | | 0 | 0.2 | 0.4 | 0.6 | 0.8 | 0.9 |
| Normal | 5 | 46.9 | 54.8 | 72.0 | 86.4 | 98.5 | 100.0 |
| | 1000 | 87.6 | 91.0 | 97.7 | 99.9 | 100.0 | 100.0 |
| Binomial | 5 | 89.8 | 94.7 | 98.4 | 99.4 | 99.9 | 100.0 |
| | 1000 | 78.6 | 85.0 | 92.7 | 98.2 | 99.7 | 100.0 |

# E  Additional Simulations

## E.1  On the Slider Parameter

To assess the influence of the slider parameter $\tau$ on the size and power of the proposed Zipper approach, we conduct a small simulation study using the simulation settings outlined in the manuscript. The results of this study are summarized in Tables 4–5. Notably, we observe that the empirical sizes are consistently controlled when $\tau$ falls within an appropriate range. However, selecting excessively large values of $\tau$ led to a slight inflation in size, primarily due to the poor normal approximation under the null hypothesis, as discussed in Section 2.5. Furthermore, the empirical power increases as the value of $\tau$ increases, which aligns with our expectations.

## E.2  Additional Simulation Experiments

To provide a comprehensive analysis of the empirical sizes and power across various settings, we conducted additional simulations using the simulation settings outlined in the manuscript. We explored different combinations of the sample size $n$ and dimension $p$. Moreover, we investigated a model with a heavy-tailed response by considering $Y = X\beta + \sigma_Y \varepsilon/3$, $\varepsilon \sim t_3$, where $t_3$ represents the $t$-distribution with 3 degrees of freedom. The results for empirical sizes and power with $\delta = 1$ are summarized in Tables 6 and 7. These additional findings consistently support the conclusion that the Zipper method demonstrates reliable empirical size performance and provides significant power enhancement compared to methods that utilize non-overlapping splits.

# F  Comparison between Zipper and Data Perturbation Methods

To evaluate variables with zero-importance, Rinaldo et al. [13] and Dai et al. [23] proposed a data perturbation technique, injecting independent zero-mean noises into empirical influence functions. We describe this method using our own terminology. We begin by partitioning the data randomly into $K$ folds, denoted as $\mathcal{D}_1, \ldots, \mathcal{D}_K$, ensuring equitable sizing across folds. For each fold index $k \in \{1, \ldots, K\}$, we construct estimators $f_{k,n}$ and $f_{k,n,\mathcal{S}}$ for the oracle prediction functions $f$ and $f_{\mathcal{S}}$, correspondingly, utilizing data excluding fold $\mathcal{D}_k$. This method employs the following test statistic:

$$\phi_{n,\mathcal{S},\text{pert},\rho} = \frac{1}{n} \sum_{k=1}^{K} \left[ \sum_{i:Z_i \in \mathcal{D}_k} \phi(Z_i) - \phi_{\mathcal{S}}(Z_i) + \rho\varepsilon_i \right],$$

Table 6: Empirical sizes (in percentage) of various testing procedures.

| Model | $n$ | $p$ | Zipper | WGSC-3 | DSP-Split | WGSC-2 | DSP-Pert |
|---|---|---|---|---|---|---|---|
| | 200 | | 3.6 | 5.0 | 3.7 | 0.4 | 12.5 |
| | 500 | 5 | 2.1 | 4.2 | 4.7 | 0.0 | 9.8 |
| | 1000 | | 3.0 | 5.7 | 3.6 | 0.3 | 10.7 |
| | 200 | | 3.2 | 4.3 | 3.9 | 1.2 | 8.9 |
| Normal | 500 | 10 | 3.7 | 4.5 | 4.9 | 0.3 | 9.7 |
| | 1000 | | 2.5 | 4.0 | 4.6 | 0.3 | 9.6 |
| | 500 | 200 | 4.7 | 6.3 | 4.8 | 17.1 | 24.6 |
| | 1000 | | 5.2 | 5.1 | 5.0 | 13.6 | 25.9 |
| | 1000 | 1000 | 4.4 | 6.3 | 3.9 | 15.8 | 36.6 |
| | 200 | | 3.5 | 3.4 | 3.7 | 0.3 | 11.0 |
| | 500 | 5 | 4.3 | 4.2 | 3.8 | 0.3 | 8.5 |
| | 1000 | | 4.3 | 3.7 | 4.6 | 0.3 | 8.1 |
| | 200 | | 2.4 | 2.6 | 2.6 | 1.3 | 8.5 |
| $t_3$ | 500 | 10 | 5.6 | 4.1 | 3.7 | 0.0 | 8.4 |
| | 1000 | | 4.3 | 4.0 | 3.6 | 0.2 | 8.6 |
| | 500 | 200 | 4.9 | 3.6 | 5.0 | 15.4 | 24.7 |
| | 1000 | | 5.4 | 3.9 | 5.3 | 12.1 | 27.8 |
| | 1000 | 1000 | 6.4 | 3.5 | 3.9 | 14.9 | 34.0 |
| | 200 | | 3.7 | 4.9 | 3.4 | 0.4 | 5.9 |
| | 500 | 5 | 3.3 | 4.5 | 3.9 | 0.2 | 4.6 |
| | 1000 | | 2.8 | 4.1 | 3.7 | 0.2 | 4.4 |
| | 200 | | 3.1 | 6.4 | 3.1 | 2.0 | 6.5 |
| Binomial | 500 | 10 | 2.8 | 5.2 | 4.1 | 0.6 | 5.0 |
| | 1000 | | 3.2 | 4.5 | 3.7 | 0.7 | 4.1 |
| | 500 | 200 | 7.0 | 3.2 | 6.3 | 15.2 | 27.9 |
| | 1000 | | 6.3 | 3.8 | 6.0 | 16.4 | 24.9 |
| | 1000 | 1000 | 6.3 | 4.2 | 5.4 | 16.9 | 32.2 |

Table 7: Empirical power (in percentage) of various testing procedures for $\delta = 1$.

| Model | | Normal | | | $t_3$ | | | Binomial | | |
|---|---|---|---|---|---|---|---|---|---|---|
| $n$ | $p$ | Zipper | WGSC-3 | DSP-Split | Zipper | WGSC-3 | DSP-Split | Zipper | WGSC-3 | DSP-Split |
| 200 | | 9.5 | 7.7 | 6.7 | 26.5 | 13.2 | 14.0 | 6.9 | 6.4 | 5.3 |
| 500 | 5 | 44.1 | 10.5 | 9.6 | 66.3 | 17.6 | 16.4 | 49.0 | 12.4 | 11.5 |
| 1000 | | 86.8 | 13.9 | 14.9 | 89.0 | 21.1 | 18.4 | 91.7 | 20.6 | 19.6 |
| 200 | | 10.3 | 7.6 | 7.9 | 28.6 | 11.4 | 13.9 | 3.7 | 7.4 | 2.5 |
| 500 | 10 | 42.0 | 11.2 | 10.4 | 68.7 | 18.4 | 18.8 | 36.2 | 12.7 | 10.3 |
| 1000 | | 87.9 | 15.7 | 11.6 | 87.6 | 18.4 | 20.5 | 87.9 | 19.5 | 16.0 |
| 500 | 200 | 63.6 | 16.7 | 14.2 | 87.0 | 34.9 | 32.8 | 79.2 | 24.9 | 23.0 |
| 1000 | | 96.5 | 27.3 | 27.0 | 93.2 | 42.1 | 42.8 | 99.1 | 49.1 | 44.8 |
| 1000 | 1000 | 10.4 | 7.7 | 5.6 | 22.5 | 6.2 | 7.8 | 22.0 | 6.7 | 8.4 |

where $\varepsilon_i \sim N(0,1)$ for $i = 1, \ldots, n$ are independent noise, and $\rho$ represents the perturbation parameter governing the extent of perturbation. It can be established that for any $\rho > 0$, $n^{1/2}(\phi_{n,\mathcal{S},\mathrm{pert},\rho} - \psi_{\mathcal{S}}) \xrightarrow{d} N(0, \nu^2_{\mathcal{S},\mathrm{pert},\rho})$, as $n \to \infty$, where $\nu^2_{\mathcal{S},\mathrm{pert},\rho} = \eta^2_{\mathcal{S}} + \rho^2$, and $\eta^2_{\mathcal{S}} = E[\{\phi(Z) - \phi_{\mathcal{S}}(Z)\}^2]$. Following the plug-in principle, we employ the normalized test statistic

$$T_{\mathrm{pert},\rho} := n^{1/2}\phi_{n,\mathcal{S},\mathrm{pert},\rho}/\nu_{n,\mathcal{S},\mathrm{pert},\rho},$$

which converges in distribution to $N(0,1)$ under $H_0$ for any $\rho > 0$. Here $\nu^2_{n,\mathcal{S},\mathrm{pert},\rho} = K^{-1}\sum_{k=1}^{K}\eta^2_{k,n,\mathcal{S}} + \rho^2$, where for each $k \in \{1, \ldots, K\}$, $\eta^2_{k,n,\mathcal{S}} = n_k^{-1}\sum_{i:Z_i \in \mathcal{D}_k}\{\dot{\mathbb{C}}(f_{k,n}, P_{k,n}; \delta_{Z_i} - P_{k,n}) - \dot{\mathbb{C}}(f_{k,n,\mathcal{S}}, P_{k,n}; \delta_{Z_i} - P_{k,n})\}^2$. For a prespecified significance level $\alpha \in (0,1)$, this method rejects $H_0$ if $T_{\mathrm{pert},\rho} > z_{1-\alpha}$.

Indeed, a direct correspondence emerges between the slider parameter $\tau$ within our Zipper and the perturbation parameter $\rho$ employed in the data perturbation technique, thereby facilitating a comparative analysis of the two methodologies. To elucidate this connection, we demonstrate that, under $H_1 : \psi_{\mathcal{S}} > 0$, the power function corresponding to the data perturbation method is

$$\Pr(T_{\mathrm{pert},\rho} > z_{1-\alpha} \mid H_1)$$
$$= \Pr\left(\frac{n^{1/2}(\phi_{n,\mathcal{S},\mathrm{pert},\rho} - \psi_{\mathcal{S}})}{\nu_{\mathcal{S},\mathrm{pert},\rho}} > z_{1-\alpha} - \frac{n^{1/2}\psi_{\mathcal{S}}}{\nu_{\mathcal{S},\mathrm{pert},\rho}} \mid H_1\right)$$
$$= \Phi\left(-z_{1-\alpha} + \frac{n^{1/2}\psi_{\mathcal{S}}}{\nu_{\mathcal{S},\mathrm{pert},\rho}}\right) + o(1)$$
$$:= G_{\mathcal{S},n,\alpha,\mathrm{pert}}(\rho) + o(1).$$

First notice that our Zipper method with $\tau = 0$ (i.e., the vanilla sample-splitting) has approximate power

$$G_{\mathcal{S},n,\alpha}(0) = \Phi\left(-z_{1-\alpha} + \frac{(n/2)^{1/2}\psi_{\mathcal{S}}}{(\sigma^2 + \sigma_{\mathcal{S}}^2)^{1/2}}\right).$$

We establish an upper bound for the perturbation parameter $\rho$, namely, $\rho^2 \le 2(\sigma^2 + \sigma_{\mathcal{S}}^2) - \eta_{\mathcal{S}}^2$, where equality yields $G_{\mathcal{S},n,\alpha,\mathrm{pert}}(\rho) = G_{\mathcal{S},n,\alpha}(0)$. Should $\rho^2 > 2(\sigma^2 + \sigma_{\mathcal{S}}^2) - \eta_{\mathcal{S}}^2$, it becomes evident that $G_{\mathcal{S},n,\alpha,\mathrm{pert}}(\rho) \le G_{\mathcal{S},n,\alpha}(0)$, suggesting that the data perturbation technique may even exhibit lower power.

The idea is to establish a relationship between $\tau$ and $\rho$ such that both methods yield similar power, given their valid size control for fixed $\tau$ and $\rho$. Employing a consistent variance estimator as per Remark 2.9, our Zipper method exhibits an approximate power of

$$G_{\mathcal{S},n,\alpha}^{(\mathrm{CI})}(\tau) = \Phi\left(-z_{1-\alpha} + \frac{\{n/(2-\tau)\}^{1/2}\psi_{\mathcal{S}}}{\nu_{\mathcal{S},\tau}}\right).$$

By setting $G_{\mathcal{S},n,\alpha}^{(\mathrm{CI})}(\tau) = G_{\mathcal{S},n,\alpha,\mathrm{pert}}(\rho)$, we obtain

$$\rho^2 = (2 - \tau)(1 - \tau)(\sigma^2 + \sigma_{\mathcal{S}}^2) - (1 - \tau)^2\eta_{\mathcal{S}}^2, \tag{8}$$

establishing a one-to-one correspondence. Under this correspondence,

$$G_{\mathcal{S},n,\alpha,\mathrm{pert}}(\rho) = G_{\mathcal{S},n,\alpha}^{(\mathrm{CI})}(\tau) \le G_{\mathcal{S},n,\alpha}(\tau),$$

where the latter denotes the approximate power of our Zipper method utilizing the proposed variance estimation scheme. Thus, our Zipper method can be regarded as a data-adaptive perturbation strategy that circumvents external randomization while potentially offering power enhancement owing to the variance estimation scheme.

Figure 4 illustrates the empirical power comparison between Zipper and the data perturbation method, where the perturbation parameter $\rho = \rho(\tau)$ is determined by (8), across various values of $\tau$. This assessment is conducted according to the low-dimensional normal response scenario outlined in Section 3, particularly with $p = 5$ and $n = 500$. As anticipated, Zipper and the perturbation method demonstrate the same power when $\tau = 0$, while for $\tau \in (0, 1)$, Zipper consistently exhibits superior power compared to the perturbation method.

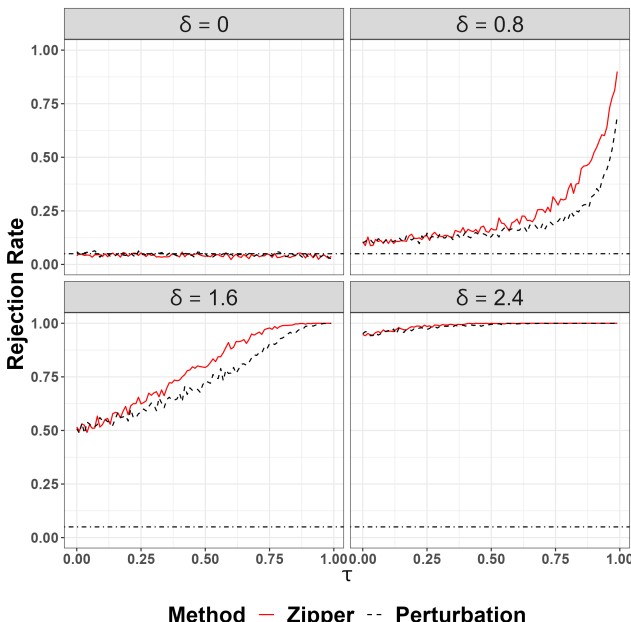

Figure 4: Empirical size and power comparison of Zipper and data perturbation method as a function of $\tau$. The dot-dashed horizontal line represents the intercept at $\alpha = 5\%$.

