# OpenReview forum: "Zipper: Addressing Degeneracy in Algorithm-Agnostic Inference"
_NeurIPS.cc/2024/Conference — NeurIPS 2024 spotlight_

### Official Review · Reviewer_DjJu · 2024-07-12

**Soundness:** 3
**Presentation:** 2
**Contribution:** 2
**Rating:** 6
**Confidence:** 2

**Summary:**

This paper proposes a method for quantifying model-agnostic goodness of fit to allow better comparison between two models / model classes etc. In particular, this paper deals with the problem of degeneracy under the null hypothesis of equal goodness. Prior work has addressed this by splitting the test set into distinct subsets, however, this reduces the sample size significantly. To overcome this, Zipper splits the test set into overlapping test sets and uses the proportion of overlap of the splits to better aggregate the test statistic. This way, Zipper is able to more effectively use limited test data.

**Strengths:**

1. The proposed method's idea seems sound and useful. Test data is indeed limited in practice and a method like Zipper can enable GoF evaluation with limited test data.

**Weaknesses:**

1. I'm not an expert in this area, but I'm concerned about the novelty and significance of this contribution. I would urge the authors to further clarify the significance of this change in test data splitting (especially for other expert reviewers).

**Questions:**

N/A

**Limitations:**

Novelty / Contribution.

---

> ### Author Rebuttal · Authors · 2024-08-05
>
> We express our sincere gratitude for your dedicated time and thoughtful review of our paper. We would like to further elucidate the novelty and significance of our contribution as follows:
>
> **Broad applicability and compatibility with flexible training techniques:** Our proposed Zipper device is designed for wide-ranging applicability across various goodness-of-fit testing scenarios, including variable importance assessment, specification testing, and model selection. Notably, this device necessitates minimal intervention in the model training process, enhancing its practicality.
>
> **Efficient utilization of test data:** Unlike the approach by Williamson et al., which divides the test data into two non-overlapping parts (thereby halving the sample size for testing), our overlap scheme in the Zipper device allows for a more efficient use of test data. This results in substantial power improvements while maintaining valid size control, as demonstrated in our theoretical results and finite-sample experiments across various contexts.
>
> We hope these points help to clarify the novelty and significance of our work. We welcome any further comments or questions you may have.

---

> > ### Comment · Reviewer_DjJu · 2024-08-13
> >
> > I have read the rebuttal and in the light of the other reviewer's comments, I change my score to recommend acceptance for this paper.

---

### Official Review · Reviewer_2sMb · 2024-07-13

**Soundness:** 3
**Presentation:** 4
**Contribution:** 3
**Rating:** 7
**Confidence:** 3

**Summary:**

The authors propose a new test statistic they call Zipper for algorithm-agnostic inference on goodness-of-fit testing. While previous solutions suffer from a degeneracy issue (i.e., fails to converge to a non-degenerate distribution under the null hypothesis of equal goodness), the proposed test statistic does not (i.e., it converges asymptotically to a normal distribution under the null). It also improves power over other proposed solutions which also tackle the degeneracy issue (due to Zipper’s effective reuse of data). Zipper could have applications in specification testing, model selection and variable importance assessments.

**Strengths:**

1. The paper is very well-written and the motivation is clear.
2. The benefit of the proposed test statistic over predecessors is clear from the theoretical analysis.
3. The algorithm for Zipper is also quite straightforward, lending to its practicality.

**Weaknesses:**

1. The paper could benefit from being more self-contained. For instance:
- The theorems reference conditions in the appendix. While I understand why it makes sense to place such details in the appendix, some informal discussion of the conditions in the main paper would be helpful for exposition.
- The paper consistently references WIlliamson et al. without precisely describing what is proposed in the work (beyond the a high-line mention of “Williamson et al. proposed an additional split of the testing data” in lines 97-98). Adding more discussion up front would help with readability / appreciating the paper’s contribution during the analysis.
2. The paper mentions that this test statistic / goodness-of-fit testing broadly is applicable in many settings but does not compare what is proposed with other methods which are also used in these settings (e.g., other measures of variable importance). It would help to add this discussion to the related work.
3. The numerical experiments / results could be more comprehensive. Specifically:
- The results for the numerical experiments do not make it easy to see that Zipper is preferred over the other methods; for instance, Table 1 shows that Zipper has better size than WSGC-1 and DSP-Pert, but Figure 2 shows that the latter have better power.
- Table 2 is confusing given the caption mentions both size and power but I only see one number for each method. I think based on the text and inferring from the magnitudes that the first row is size while the second and third row are power, but why not include both for all scenarios?
- Based on Figure 3, it is not obvious to me that the left column (Zipper) is better than the right column (baseline). Could the authors elaborate?
- The text in 3.2.2 makes it more clear what might be preferable about Zipper, but it is not clear based on Table 3 alone. Perhaps some way of directly showcasing the alignment with a previous study would be helpful?

**Questions:**

1. Lines 231-236 suggest that it is not only the zipper mechanism that enables power improvements, but also the use of a variance estimator. Could the authors expand on the latter, e.g. provide more intuition, incorporate it into the overall story in the paper outside of these few lines?
2. When would one prefer this approach to others in, say, variable importance testing (e.g. conditional randomization tests, Shapley values) and why? I understand that the precise goal is slightly different between the options mentioned, but as this work falls under the interpretability / explainability category, it would be useful to situate it explicitly.
3. Could the authors combine the results of Table 1 and Figure 2 to help visualize how Zipper compares holistically to the other methods?
4. Could the authors provide both power and size results in Table 2?
5. Could the authors explain why Figure 3 shows that the proposed method is preferred over the baseline?

**Limitations:**

The applicability of the method for larger scale settings has not yet been tested. Moreover, for an application such as explanability, GoF testing may not necessarily be the most direct given the test merely tests how well a pre-specified subset performs relative to the full set, rather than locating an important subset.

(As a quick aside, I noticed in the checklist that the authors mention discussing limitations in Section 4, but I do not seem to see any mentions of limitations in that section)

---

> ### Author Rebuttal · Authors · 2024-08-05
>
> Thank you for your detailed review and constructive feedback on our paper. Your insights are invaluable for refining our manuscript.
>
> **Weaknesses:**
>
> 1. _Self-contained discussions:_
>
> - _On conditions:_ Condition (C1) pertains to the optimality of the prediction function $f$, eliminating first-order estimation biases. Condition (C2) requires Hadamard differentiability of the predictiveness criterion. Condition (C3) demands accurate estimation of $f$, rendering second-order terms negligible. Condition (C4) controls the remainder terms. Condition (C5) ensures consistent variance estimators. We will add a detailed discussion of these conditions in the main text to improve readability.
>
> - _On Williamson et al.:_ Williamson et al. introduced a framework for algorithm-agnostic variable importance assessments using cross-fitting and demonstrated the asymptotic linearity of the test statistic. They also advocated additional splitting of the test data for null importance testing. We extend this framework to accommodate a wider range of goodness-of-fit tests, including specification testing and model selection. Our method, Zipper, promotes data reuse, significantly augmenting the testing power. We will elaborate on these points in the Related Work section to strengthen narrative coherence.
>
> 2. _Related work:_ For variable importance assessments, in addition to LOCO methods within our framework, Shapley value-based measures are commonly used (Lundberg and Lee, 2017; Covert et al., 2020, 2021; Kumar et al., 2020). These measures, which estimate the incremental predictive accuracy contributed by a specific variable across all possible covariate subsets, reveal complex inter-variable relationships but at a considerable computational expense. Furthermore, conditional randomization tests (Candes et al., 2018; Tansey et al., 2021) offer a robust alternative when covariate distributions are known or can be accurately estimated. These methods are especially beneficial in semi-supervised settings with extensive unlabeled data. Additional methods, such as LIME (Ribeiro et al., 2016) and Floodgate (Zhang and Janson, 2020), will be discussed to expand the comprehensiveness of our manuscript's Related Work section.
>
> 3. _Comprehensiveness of numerical experiments:_
>
> - _Preference of Zipper in Figure 2:_ Despite WSGC-2 and DSP-Pert showing superior power in certain settings ($p=1000$), both exhibit considerable size inflation-approximately 0.2 and 0.4 respectively, where the nominal level is 0.05. This size distortion, depicted at $\delta=0$ in Figure 2, renders these methods less reliable for practical application. Conversely, Zipper maintains robust size control and achieves high power, affirming its utility.
>
> - _Size and power in Table 2:_ To clarify, Scenario (i) represents a specific case of $H_0$, while Scenarios (ii) and (iii) exemplify $H_1$ instances. We will rename Scenario (i) to $H_0^{'}$ and Scenarios (ii) and (iii) to $H_1^{'}$ and $H_1^{''}$, respectively.
>
> - _Improvement over baseline in Figure 3:_ Both Zipper and the baseline method (WGSC-3) demonstrate effective size control. In the MNIST handwriting dataset example, we perform sequential variable importance tests to determine the relevance of each region for prediction. We apply Bonferroni corrections to address multiple testing concerns. Figure 3 indicates that Zipper identifies more significant regions (five regions filled in red) compared to the baseline (two regions).
>
> - _Alignment with a previous study in Table 3:_ In our analysis of the Bodyfat dataset, we perform multiple tests with a Bonferroni correction set at $\alpha=0.05/10=0.005$. Table 3 reveals that Zipper successfully identifies both the Abdomen and Hip as significant variables (both $\le 0.005$), whereas WGSC-3 identifies only the Abdomen. Notice that the detection of the Hip variable corroborates results from previous research. We will highlight these outcomes in the revised manuscript.
>
> **Questions**:
>
> 1. _Variance estimator:_ The improvement in testing power is attributed to both the overlapping mechanism of Zipper and the variance estimator. This estimator is consistent under $H_0$, thereby ensuring valid size control. Under $H_1$, it underestimates the actual unknown variance, which is advantageous for power enhancement as demonstrated. However, for confidence interval construction-e dual problem of hypothesis testing-a consistent variance estimator under both $H_0$ and $H_1$ is necessary. We provide such an estimator. See Remark 2.9 and Section D.
>
> 2. _Preference of Zipper in variable importance testing:_ This is addressed in our response to Weaknesses 2.
>
> 3. _Combination of results of Table 1 and Figure 2:_ Please refer to our response to Weaknesses 3, Point 1.
>
> 4. _Power and Size in Table 2:_ This is explained in our response to Weaknesses 3, Point 2.
>
> 5. _Explanation of Figure 3:_ Please see our response to Weaknesses 3, Point 3.
>
> **Limitations** in _larger scale settings and locating important subsets_: We acknowledge and value this critical feedback. Zipper is specifically designed to evaluate the significance of individual features or subsets in predictions, which paves the way for large-scale comparisons as mentioned at the end of Section 4. Notably, we can conduct a sequence of variable importance tests, each aimed at assessing the relevance of a specific variable $X_j$ in the predictive model while controlling for a global error rate. This procedure necessitates the fitting of $p+1$ models: one that includes all variables and $p$ null models, each excluding a distinct variable. Such a process is computationally demanding. Moreover, accurately controlling error rates presents a considerable challenge due to complex dependency structures among the p-values. We will elucidate this point further in the revised manuscript.
>
> We appreciate the opportunity to enhance our paper based on your feedback and look forward to further discussions.
>
> ####

---

> > ### Comment · Reviewer_2sMb · 2024-08-07
> >
> > Thank you for the comprehensive response! A few follow-up questions:
> > - For W2, could the authors provide more discussion one when to consider this class of methods over the others added to the related works discussion?
> > - For your response Q1, you mentioned, "Under $H_1$, it (the variance estimator) underestimates the actual unknown variance, which is advantageous for power enhancement as demonstrated." Could you explain why that is and where it is demonstrated?

---

> > > ### Author Response · Authors · 2024-08-09
> > > **On related works and variance underestimation**
> > >
> > > Thanks for your reply!
> > >
> > > - For W2, could the authors provide more discussion one when to consider this class of methods over the others added to the related works discussion?
> > >
> > > Response:
> > >
> > > _Shapley value-based methods_: Shapley value-based measures can be conceptualized as a weighted average of LOCO measures within our framework, accounting for the inclusion of a feature (or a feature group) of interest across all possible subsets (Williamson and Feng, 2020; Verdinelli and Wasserman, 2023; 2024). Inference based on Shapley values can utilize sample-splitting methods, as demonstrated by Williamson and Feng (2020). While extending our Zipper framework to accommodate this sample-splitting inference is feasible, it requires the evaluation of all covariate subsets, entailing significant computational resources due to the necessity of fitting numerous models. Moreover, as noted by Verdinelli and Wasserman (2024), many of these submodels may not contribute meaningfully to the overall measure, particularly when each is weighted equally in the definition. We recommend that if there is confidence in the algorithms' ability to approximate true submodels accurately, LOCO measures should be considered, where our framework can be effectively implemented.
> > >
> > > _Conditional randomization tests_: CRTs offer flexibility in selecting variable importance measures beyond LOCOs. This method necessitates knowledge of the conditional distribution $X_\mathcal{S}\mid X_{-\mathcal{S}}$, from which data can be sampled for multiple model refitting and measure calculation (Candes et al., 2018). Subsequent research, such as that by Tansey et al. (2021), has aimed to mitigate these computational demands. However, when the distribution of covariates is either unknown or cannot be precisely estimated, our methodology presents a more viable alternative to CRTs.
> > >
> > > - For your response Q1, you mentioned, "Under H1, it (the variance estimator) underestimates the actual unknown variance, which is advantageous for power enhancement as demonstrated." Could you explain why that is and where it is demonstrated?
> > >
> > > Response:
> > >
> > > _Variance underestimation under $H_1$_: As outlined in Lines 167 and 218, the unknown variance $\nu_{\mathcal{S},\tau}^2=(\nu_{\mathcal{S},\tau}^{(0)})^2+\tau\eta^2_\mathcal{S}$, where $(\nu_{\mathcal{S},\tau}^{(0)})^2=(1-\tau)(\sigma^2+\sigma_{\mathcal{S}}^2)$. Given that $\eta^2_\mathcal{S}\ge 0$ (with $\eta^2_\mathcal{S}=0$ under $H_0$ and $\eta^2_\mathcal{S}>0$ under $H_1$), it follows that $\nu_{\mathcal{S},\tau}^2\ge(\nu_{\mathcal{S},\tau}^{(0)})^2$. Proposition 2.3 and Line 221 confirm that our variance estimator is always consistent to $(\nu_{\mathcal{S},\tau}^{(0)})^2$ under both hypotheses. Thus, under $H_1$, the estimator tends to underestimate $\nu_{\mathcal{S},\tau}^2$.
> > >
> > > _Power enhancement_: The aforementioned underestimation leads to an advantageous adjustment in our approximated power function, as outlined in Theorem 2.6 and Line 232. Specifically,
> > >
> > > $$
> > > G_{\mathcal{S},n,\alpha}(\tau) = \Phi\left(-\frac{\nu_{\mathcal{S},\tau}^{(0)}}{\nu_{\mathcal{S},\tau}}z_{1-\alpha} + \frac{\{n/(2-\tau)\}^{1/2}\psi_\mathcal{S}}{\nu_{\mathcal{S},\tau}}\right)\ge \Phi\left(-z_{1-\alpha} + \frac{\{n/(2-\tau)\}^{1/2}\psi_\mathcal{S}}{\nu_{\mathcal{S},\tau}}\right),
> > > $$
> > >
> > > This expression establishes a lower bound that corresponds to the power function of a test statistic employing a consistent variance estimator for $\nu_{\mathcal{S},\tau}^2$ under both $H_0$ and $H_1$. This explains how the underestimation enhance the test's power. Between Lines 231-236, we further elucidate that this lower bound outperforms the power achievable by conventional sample-splitting-based statistics, thereby underscoring the efficacy of our overlapping mechanism in boosting power.
> > >
> > > Thank you once again for your insightful questions.
> > >
> > > References
> > >
> > > Brian Williamson and Jean Feng. Efficient nonparametric statistical inference on population feature importance using Shapley values. *Proceedings of the 37th International Conference on Machine Learning*, PMLR 119:10282-10291, 2020.
> > >
> > > Isabella Verdinelli and Larry Wasserman. Feature importance: A closer look at shapley values and loco. *arXiv:2303.05981*, 2023.
> > >
> > > Isabella Verdinelli and Larry Wasserman. Decorrelated variable importance. *Journal of Machine Learning Research*, 25(7):1–27, 2024.
> > >
> > > Emmanuel Candes, Yingying Fan, Lucas Janson, and Jinchi Lv. Panning for gold:‘model-x’knockoffs for high dimensional controlled variable selection. *Journal of the Royal Statistical Society: Series B (Statistical Methodology)*, 80(3):551–577, 2018.
> > >
> > > Wesley Tansey, Victor Veitch, Haoran Zhang, Raul Rabadan, and David M Blei. The holdout randomization test for feature selection in black box models. *Journal of Computational and Graphical Statistics*, 31(1):151–162, 2022.

---

### Official Review · Reviewer_aa86 · 2024-07-18

**Soundness:** 4
**Presentation:** 4
**Contribution:** 4
**Rating:** 8
**Confidence:** 4

**Summary:**

This work presents a test of Performance(f, test_data_1) - Performance(f_subset, test_data_2) where f is the best model in a class F and f_subset is the best model in a subset of F. They focus on how precisely to split your samples between test_data_1 and test_data_2 to avoid degeneracies that can arise when asking questions like whether the performance difference is equal to zero.

The main proposed solution is to take an underlying set of test data D_k, split it into two possibly overlapping pieces D_A and D_B, with overlap D_o, and to compute Performance(f, test_data_1) as a combination of performance on D_o and D_A \ D_o, and likewise to compute Performance(f_subset, test_data_2) as a combination of performance on D_o and D_B \ D_o.

For example, Performance(f, test_data_1) = \tau performance(f, D_o) + (1-\tau) performance(f, D_A \ D_o) . Here picking \tau 0<=\tau<1 avoids the degeneracy.  This idea can be seen as executing a particularly careful and favorable property of sample splitting.

They show that the test statistic, under the null hypothesis of no difference, is asymptotically linear. They also derive a consistent estimator for the variance of the statistic under the null.  They also provide selection criteria for the zipper \tau.

Using these tools they analyze the power/size of the test statistic and run simulations in variable importance, model specification, and more.

As future work, they leave open some ideas around which other hyperparameters to select data-adaptively, how the test works on large scale data, and how to control errors.

**Strengths:**

- broadly applicable scenario (model/algorithm agnostic inference)
- solves a known issue of a popular test statistic under the null hypothesis of 0 performance difference
- does so with minimal intervention to model fitting / testing (just split your data in a certain way, evaluate on carefully defined subsets, and combine)
- offers guidance on a few hypeparameters
- provides results including on estimating the variance of the test statistic.
- suggests fruitful directions for future work.

**Weaknesses:**

Nothing major at the moment, may revise before discussion period.

**Questions:**

Nothing major at the moment, may revise before discussion period.

**Limitations:**

Yes

---

> ### Author Rebuttal · Authors · 2024-08-05
>
> We extend our sincere gratitude for your thoughtful and detailed review of our paper. We greatly appreciate your positive evaluation and constructive feedback. We are committed to further refining our manuscript and eagerly welcome any additional comments or questions you may have.

---

### Decision · Program_Chairs · 2024-09-25

**Decision:**

Accept (spotlight)

**Comment:**

The paper Addresses degeneracy issues in algorithm/model-agnostic goodness-of-fit inference such as assessing variable importance.
It broadly applicable scenario (model/algorithm agnostic inference).
Also, it does so with minimal intervention to model fitting / testing (just split your data in a certain way, evaluate on carefully defined subsets, and combine).
The authors clearly addressed all questions by the reviewers during the rebuttal period please update the paper with the clarifications discussed.